

# Turbulent and non-turbulent exchange of scalars between the forest and the atmosphere at night in Amazonia

Pablo E. S. Oliveira[1], Otávio C. Acevedo[1], Matthias Sörgel[2], Anywhere Tsokankunku[2], Stefan Wolff[2], Alessandro C. Araújo[3], Marta O. Sá[4], Antônio O. Manzi[4], and Meinrat O. Andreae[2]

[1]Departamento de Física, Universidade Federal de Santa Maria, Av. Roraima 1000, Santa Maria, RS, Brazil
[2]Biogeochemistry Department, Max Planck Institute for Chemistry, P.O. Box 3060, 55020 Mainz, Germany
[3]Empresa Brasileira de Pesquisa Agropecuária (EMBRAPA), Trav. Dr. Enéas Pinheiro, Belém, PA, Brazil
[4]Instituto Nacional de Pesquisas da Amazônia (INPA), Av. André Araújo 2936, Manaus, AM, Brazil

*Correspondence to:* Pablo E. S. Oliveira (pablo@ufsm.br)

**Abstract.** Nocturnal turbulent kinetic energy (TKE) and fluxes of energy, $CO_2$ and $O_3$ between the Amazon forest and the atmosphere are evaluated for a 20-day campaign at the Amazon Tall Tower Observatory (ATTO) site. The distinction of these quantities between fully turbulent (weakly stable) and intermittent (very stable) nights is discussed. Spectral analysis indicates that low-frequency, non-turbulent fluctuations are responsible for a large portion of the variability observed on intermittent
nights. In these conditions, the low-frequency exchange may dominate over the turbulent transfer. In particular, we show that within the canopy most of the exchange of $CO_2$ and $H_2O$ happens on temporal scales longer than 100 s. At 80 m, on the other hand, the turbulent fluxes are almost absent in such very stable conditions, suggesting a boundary layer shallower than 80 m. The relationship between TKE and mean winds shows that the stable boundary layer switches from the very stable to the weakly stable regime during intermittent bursts of turbulence. In general, fluxes estimated with long temporal windows that
account for the low-frequency effects are more dependent on the stability over a deeper layer above the forest than they are on the stability between the top of the canopy and its interior, suggesting that low-frequency processes are controlled over a deeper layer above the forest.

## 1  Introduction

The turbulence structure above forested canopies has been an important subject of research over the past decades. Such knowl-
edge is essential to answer relevant scientific questions such as the quantification of the exchange of scalars between forested ecosystems and the atmosphere. Some of the precursor studies in this field have been performed in the Amazon region during projects such as ABLE 2A and 2B (Fitzjarrald et al., 1988; Garstang et al., 1990; Fan et al., 1990). Subsequent projects in this region that kept the focus on this subject include ABRACOS (Grace et al., 1995; Malhi et al., 1998; Kruijt et al., 2000), LBA (Araújo et al., 2002; Saleska et al., 2003; Miller et al., 2004), GO-Amazon (Fuentes et al., 2016; Santos et al., 2016) and, most
recently, ATTO (Andreae et al., 2015; Zahn et al., 2016).

Ultimately, one of the most relevant questions that these projects aimed to answer is the role of the Amazon rainforest as either a net sink or source of $CO_2$ to the atmosphere. Results diverge largely among the studies, from a net sink of 5.9 T C





$ha^{-1}$ $yr^{-1}$ found by Grace et al. (1995) to a net source of 1.3 T C $ha^{-1}$ $yr^{-1}$ found by Saleska et al. (2003). Although some of this variability may be accepted as genuine, caused by site or interannual differences, it is now well established that methodological problems had affected the estimates that found enhanced carbon uptake. Most of these issues regard the treatment of nocturnal data, as a consequence of the complex nature of the atmospheric flow during night under stable conditions. In particular, during very stable nights, turbulent mixing is reduced and constrained to small temporal scales (Vickers and Mahrt, 2006; Acevedo et al., 2014). The exchange of properties such as $CO_2$ from the forest to the atmosphere may occur mostly through non-turbulent motion, such as drainage flows (Staebler and Fitzjarrald, 2004; Aubinet et al., 2003; Feigenwinter et al., 2004; Tóta et al., 2008) or by transport on temporal scales longer than those that characterize the turbulent flow (Santos et al., 2016).

The motion with temporal fluctuations longer than turbulence but smaller than those produced by mesoscale systems has been referred to as "submeso" by Mahrt (2009) and it has become an important subject of micrometeorological research since then. Typically, these non-turbulent fluctuations may be larger in magnitude than their turbulent counterpart, and they may introduce fluxes that are larger as well. On the other hand, these fluxes are not driven by local gradients, so that they are also much more variable than the turbulent fluxes, and of either sign, in such a way that their overall contribution frequently averages out over longer periods (Vickers and Mahrt, 2003). Nevertheless, their contribution may be important for closing the budgets over smaller time periods (Acevedo and Mahrt, 2010; Kidston et al., 2010).

Many studies on turbulence above and within forested canopies have presented an analysis of the spectral distribution of the turbulence velocity components and of their vertical variation with respect to the canopy top (Baldocchi and Meyers, 1988; Blanken et al., 1998; Dupont and Patton, 2012). In general, these studies focused on the time scale of the turbulent exchange and how it depends on factors such as distance from canopy top and atmospheric stability. Equivalent analyses focusing on scalar flux cospectra have not been presented as often. Sakai et al. (2001) and Finnigan et al. (2003) used cospectral similarity to conclude that low-frequency contribution could account for missing energy and $CO_2$ fluxes in their respective budgets, but neither study addressed how the cospectra varied across the canopy. Other studies (Campos et al., 2009; Fares et al., 2014) looked at scalar flux cospectra with the specific purpose of identifying the proper temporal scale for turbulent flux determination. Santos et al. (2016) found that horizontal turbulent kinetic energy (TKE) spectra are bimodal above an Amazonian rain forest canopy, with the peak at short time scales being related to turbulence and the peak at longer time scales being associated with non-turbulent, submeso fluctuations. Within the canopy, on the other hand, only the peak at longer time scale is preserved, indicating that non-turbulent fluctuations above the canopy propagate downward more efficiently than the turbulent ones. They also found that sensible heat flux cospectra within the canopy peaked at longer time scales, again similar to those of the non-turbulent maxima of horizontal TKE above the canopy. This result indicates that the exchange of scalars between the canopy and the atmosphere at night may occur at longer time scales than those traditionally used in the eddy covariance approach. Their study, however, did not include the analysis of $CO_2$ or latent heat fluxes and reactive trace gases like $O_3$, so that the question whether these quantities are affected by similar processes remains open.

A comparison of scalar flux cospectra within and above a forested canopy, aimed specifically at addressing the contribution of non-turbulent flow to the total fluxes at the different heights, has not been presented previously. The present study aims at



addressing this issue and to evaluate how these exchange processes affect the scalar profiles that are routinely measured at ATTO.

## 2 Data and Methods

### 2.1 Experimental site

The dataset was collected during the ATTO (Amazon Tall Tower Observatory)-IOP1 campaign in October/November 2015 at Reserva de Desenvolvimento Sustentável Uatumã (Uatumã Sustainable Development Reserve – USDR), in the Amazon region. The site is located on a plateau at 120 m above sea level, approximately 150 km northeast of Manaus and 12 km northeast of the Uatumã River. The average height of the highest trees near the tower is 37 m. Further information regarding terrain, soil and vegetation can be found at Andreae et al. (2015).

Micrometeorological observations were carried out on an 80-m walk-up tower with rectangular cross section at five different levels: 14, 22, 41, 55, and 80 m above the ground, the first two levels being within the forest canopy, while the three others are above it. Fast-response wind measurements were performed at all levels (CSAT3, Campbell Scientific Inc., at 14, 41 and 55 m, IRGASON, Campbell Scientific Inc., at 22 m, and Windmaster, Gill Instruments Limited, at 80 m). Scalar concentrations of $CO_2$ and water vapor were measured at 22 m (IRGASON, Campbell Scientific Inc.), 41 and 80 m (LI-7500A, LI-COR

Inc.). The diurnal cycle of the $H_2O$ mixing ratios at 41 m was erroneous for unidentified reasons. The short-term (up to 20 min) variations were correct, but the longer trend did neither agree with the other open path instruments at 22 m, and 80 m, a nearby psychrometer (Frankenberger type, Theodor Friedrichs GmbH, Germany), or with the profile measurements (see below). Therefore, the water vapor mixing ratios at that level haven been corrected by separating the short term fluctuations from the trend by applying a running mean with a window size of 5 min and adding this high frequency contribution to the

running mean (window size 5 min) of the nearby psychrometer. Scalar concentrations of ozone were measured at 41 m with chemiluminescence $O_3$ sondes (Enviscope, Germany). In front of the fast $O_3$ instrument there was a 5 m long 3/4 inch (7.52 mm inner diameter) Teflon tubing with a Teflon inlet filter (47 mm diameter, 5 µm pore size). The flows were varying due to filter clogging. After a filter change the flows were $21 \, l \, min^{-1}$ and $23.5 \, l \, min^{-1}$, respectively, whereas before the filter change they were $16 \, l \, min^{-1}$ and $14 \, l \, min^{-1}$, respectively. The resulting lag times were 0.6-0.95 sec and the Reynolds numbers in

the tubing were 2400 to 4000 at 35 °C. On the days considered for the case studies (14 and 15 November), the flow was about $16 \, l \, min^{-1}$ and the residence time was therefore 0.8 sec. All the data were collected at 10-Hz rate. As the signal of the fast $O_3$ sondes has a considerable drift, it was calibrated to a slow $O_3$ analyzer (TEI 49i, Thermo Scientific) as described by Zhu et al. (2015). The $CO_2$ profiles were measured sequentially by $CO_2/H_2O$ analyzers (LI-7000, LI-COR Inc.) connected to heated inlets at 8 heights (0.05 m, 0.5 m, 4.0 m, 12.0 m, 24.0 m, 38.3 m, 53.0 m and 79.3 m). During the case study nights

(November 14 and 15) only the LI-7000 after the Nafion®dryer was running and therefore the water vapor values could not be used. The $O_3$ profiles were also measured using the same inlet system with an Ozone Analyzer (TEI 49i, Thermo Scientific). Ambient air was continuously drawn from the inlets through non-transparent PTFE-tubing (3/8") to a valve block, which switched between the different inlet levels, so that one intake height was purged by the sample pump (PTFE coated) while





all the others were purged by the bypass pump. A time interval of 1 min was necessary for getting a constant and reliable signal for each concentration level: a complete cycle took 8 x 2 = 16 min, providing 2 measurements per level. Three 16-min measurement cycles plus one shorter 12-min cycle were performed every hour. During that last cycle, a small compromise was made to fit 4 cycles into the hour, and valve switches occurred every 90 s, thereby allowing for only one concentration

measurement at each level. The ambient air inlets mounted on the tower were protected from rain entering the inlet line by polyethylene funnels and from insects by polyethylene nets. A PTFE-filter (5 μm) was installed right after the inlet. The tubing was insulated with Styrofoam and heated. The internal temperature and pressure corrections of the LI-7000 were used, but to further minimize pressure effects, the air drawn from the inlets for analysis was sampled at the exit of the Teflon pump, so that the measurements were made close to ambient pressure for all measurement levels. The entire setup was comparable to the

profile system employed by Mayer et al. (2011).

## 2.2    Data Analysis

In the present study, 20 days of nocturnal data were analyzed, from 1 to 20 November 2015. To avoid sampling intense events associated with the transitional characteristics between daytime and nocturnal boundary layers, the evening period between the sunset and 20:00 local time (LT) was not considered for this analysis. For this reason, nocturnal periods were restricted from

20:00 to 05:00 (LT). Since the different levels of flow structures are analyzed simultaneously, only the data when all levels were available was used. The 14-m level frequently presented gaps and was not considered for this study. Therefore, the levels included are: one inside the canopy (22 m), one just above canopy top (41 m), and two levels well above the canopy (55 m and 80 m).

All the time series have been subject to quality control, which caused the removal of those series, which showed multiple

spikes or spectra that did not converge to zero at the highest frequencies. For any case where a given series was discarded for a given variable, it has not been used for any of the variables. With these restrictions, 15 nights were kept for the final analysis. Ozone measurements started on 11 November 2015, so that only 9 nights of ozone flux data were available.

The data were analyzed using two different time windows: 5 and 109 min. The multiresolution decomposition (Howell and Mahrt, 1997; Vickers and Mahrt, 2003; Voronovich and Kiely, 2007) was applied to 109 min, which corresponds to groups

of $2^{16}$ data points. In contrast to the Fourier transform, which determines periodicity, this technique mainly extracts the width of the dominant turbulent events by locally decomposing the variances. For this reason, the multiresolution spectrum (S) and cospectrum (C) have the property that the integration up to a given time scale t is equal to the variance and covariance, respectively, for a t-long time series. Consequently, the multiresolution value for a given time scale captures the physical processes (and the flux) whose duration is smaller than that time scale.

The multiresolution decomposition was applied sequentially to the time series, starting at 20:00 LT, with an overlap of 30 min between the subsequent series, totaling 14 decompositions for each night. A total of 200 series was used in the study, considering all nights. Variances and fluxes with a 109-min long time average were obtained from the integration of the respective multiresolution spectra and cospectra. Therefore, sensible and latent heat, $CO_2$ and ozone fluxes are given by $F_H = \sum_{\tau} C_{w\theta}$,





$F_q = \sum_\tau C_{wq}$, $F_C = \sum_\tau C_{wC}$, and $F_O = \sum_\tau C_{wO}$, turbulent kinetic energy is $TKE = 0.5 \sum_\tau (S_u + S_v + S_w)$ and the standard deviation of the vertical wind component is $\sigma_w = \sum_\tau S_w$. Other variables, such as the Richardson number (Ri) and average horizontal wind speed ($V$) were calculated using the same data series used in the multiresolution decomposition.

At night, it is expected that the temporal scales of turbulent transport are smaller. Campos et al. (2009) showed at another site in the Amazon forest that the contribution of turbulence to the nocturnal fluxes above the canopy occurs at temporal scales smaller than 200 s. The use of a 109-min long time window is necessary to determine the contribution of turbulent and non-turbulent motions to the fluxes. However, in order to attempt to reduce any contribution from non-turbulent transport, statistical moments, fluxes and other variables were also calculated using a 5-min time window, as used by Dupont and Patton (2012). Quantities such as sensible heat $\left(F_H = \overline{w'\theta'}\right)$, latent heat $\left(F_q = \overline{w'q'}\right)$, $CO_2$ $\left(F_C = \overline{w'C'}\right)$, and ozone $\left(F_O = \overline{w'O'}\right)$ fluxes, turbulent kinetic energy (TKE), the average horizontal wind speed (V), Richardson number (Ri), and $\sigma_w$ were determined for both 5 and 109 min. The turbulent velocity scale, defined as $V_{TKE} = \sqrt{TKE} = [0.5\left(\sigma_u + \sigma_v + \sigma_w\right)]^{1/2}$, was calculated for 5-min time windows only. This study comprises of a total of 1,577 5 min windows.

The bulk Richardson number was used to quantify atmospheric stability. The choice of the bulk instead of the flux Richardson number for the analysis has two reasons: to avoid self-correlation (Hicks, 1978; Klipp and Mahrt, 2004; Baas et al., 2006) and to quantify better the stability in very stable conditions, when fluxes are expected to approach zero. Similarly as used by Bosveld et al. (1999); Mammarella et al. (2007); Oliveira et al. (2013), a "within-canopy Richardson number" ($Ri_{can}$) and an "above-canopy Richardson number" ($Ri_{top}$) (Santos et al., 2016) were defined as

$$Ri_{can} = \frac{g}{\Theta}\Delta z \frac{\theta_{41m} - \theta_{22m}}{\left(V_{41m} - V_{22m}\right)^2} \tag{1}$$

and

$$Ri_{top} = \frac{g}{\Theta}\Delta z \frac{\theta_{80m} - \theta_{41m}}{\left(V_{80m} - V_{41m}\right)^2} \tag{2}$$

where g is the gravitational acceleration, $\Theta$ is the average potential temperature in the layer, $\Delta z$ is the height difference between the two levels, and $\theta$ and $V$ are the mean potential temperature and average horizontal wind speed at each level, respectively.

## 3 Case studies

The nocturnal flow at the site is characterized by the superposition of turbulent and non-turbulent fluctuations. In a fully turbulent night, such as 15 November 2015 (Fig. 1), there is a clear dominant wind direction at all levels. In this case, it is very rare that the horizontal wind components switch sign above the canopy. In contrast, during the intermittent night of 14 November 2015 (Fig. 2), there is no dominant wind direction at any level above the canopy. Low-frequency fluctuations are





superposed on the turbulent fluctuations, causing the mean wind direction to change quadrants frequently throughout the night. Such fluctuations have been recently attributed to submeso flow (Mahrt, 2009), while in the pollutant dispersion community similar phenomena are often referred to as meandering (Oettl et al., 2005). The most relevant difference between the two nights regards the magnitude of the turbulent mixing (Table 1). All relevant turbulence statistics are significantly larger on 15 November than on 14 November. The relative difference of the turbulence statistics between nights increases steadily in the vertical. As an example, TKE at 41 m is 3.4 times larger in the turbulent night than in the intermittent case, while at 80 m, TKE is 8.2 times larger in the turbulent night. Similar increases occur for the corresponding ratios of $\sigma_w$ and $u_*$ between the two nights.

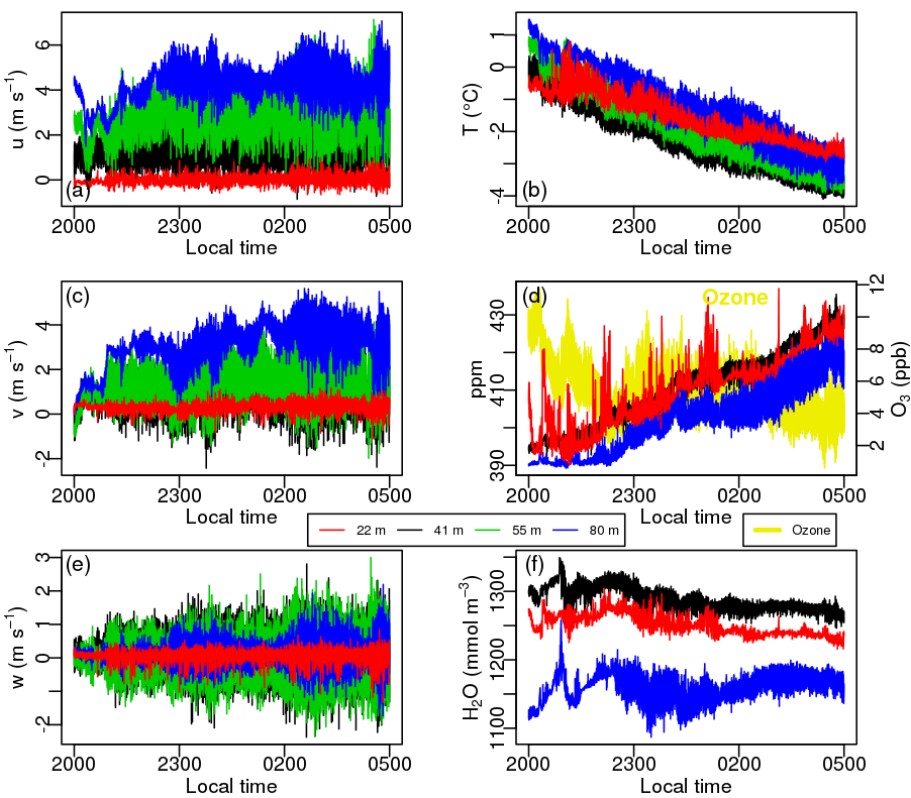

**Figure 1.** Time series of horizontal (a and c) and vertical (e) wind components, temperature perturbation from the 20:00 LT value at 41 m (b), $CO_2$ and $O_3$ (d) and water vapor (f) concentrations for the turbulent night.

Another interesting characteristic that indicates a contrast between the two nights shown in Fig. 1 and Fig. 2 regards the degree of vertical coupling across the levels, a phenomenon that has been observed by van Gorsel et al. (2011); Oliveira et al. (2013); Jocher et al. (2017). In the turbulent night, temperatures were always similar between the levels of 41 and 55 m, while at 80 m it was slightly warmer, but with the same cooling tendency throughout the period. $CO_2$ was correspondingly similar between 22 and 41 m, with the same tendencies and slightly lower values at 80 m. Although the mean trend is similar at 22 m





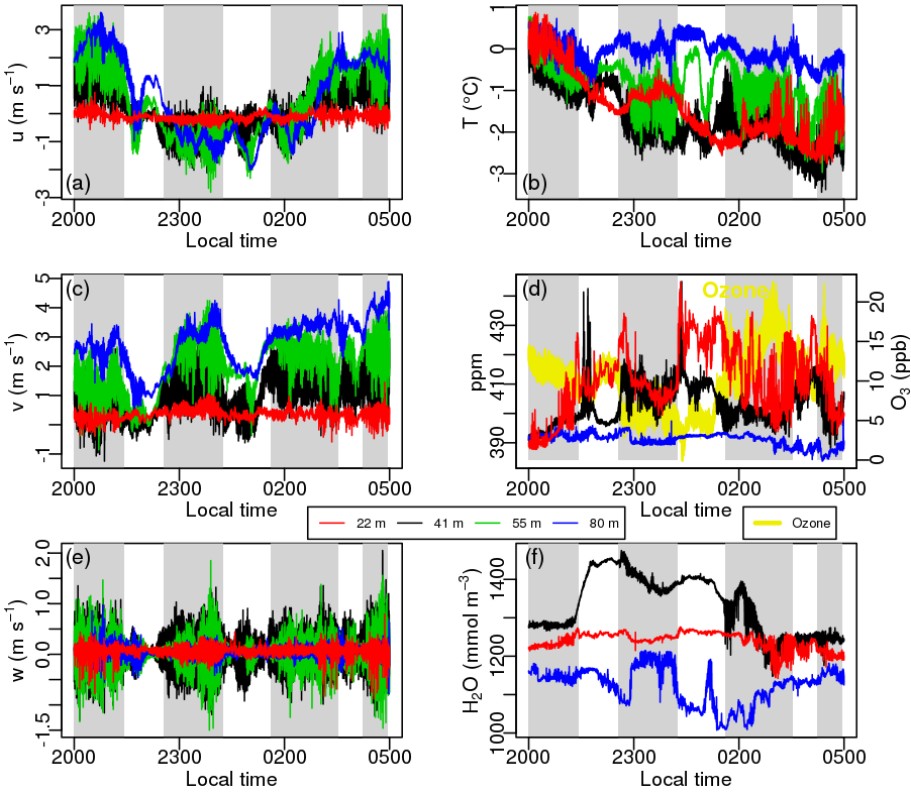

**Figure 2.** Same as in Fig. 1, but for the intermittent night. Shaded areas indicate intermittent turbulence bursts.

and 41 m, substantial short time deviations towards higher $CO_2$ values were observed at 22 m (Fig. 1d). This is in line with the higher variability of $CO_2$ values in the lower canopy, as can be seen from the profiles (Fig. 3a). This higher variability and stronger gradient (in both $CO_2$ and $O_3$) in the lower canopy point to a decoupling of the sub canopy even in the turbulent night. As the 22 m level is within the maximum of the LAI ($\sim$24 m), which separates the upper canopy from the lower canopy, it will

5   be influenced by both regimes. The gradients between 24 m and 38 m are always positive for $O_3$ and negative for $CO_2$. This can be related to the reactivity of $O_3$ as it reacts with compounds emitted from the soil (mainly NO) and plants (alkenes) and is not only taken up by stomata, but is also deposited to leaf surfaces in considerable amounts, especially under humid conditions (Fuentes and Gillespie, 1992; Rummel et al., 2007). At night, $CO_2$ is emitted by soils and plants due to respiration, causing a negative gradient.

10  All quantities showed much larger variation across the levels in the intermittent night (Fig. 2). Furthermore, sporadic events of coupling occurred during bursts of intermittent turbulence (Fig. 2, shaded areas). During these events, the gradients of temperature and $CO_2$ concentration became sporadically smaller across the vertical, except for the 80-m level, indicating that the coupling induced by the events extended over a layer shallower than 80 m. In general, the temporal evolution of all scalars show a monotonic increase (of $CO_2$) or decrease (of temperature and $O_3$) throughout the turbulent night at all levels (Fig. 1).





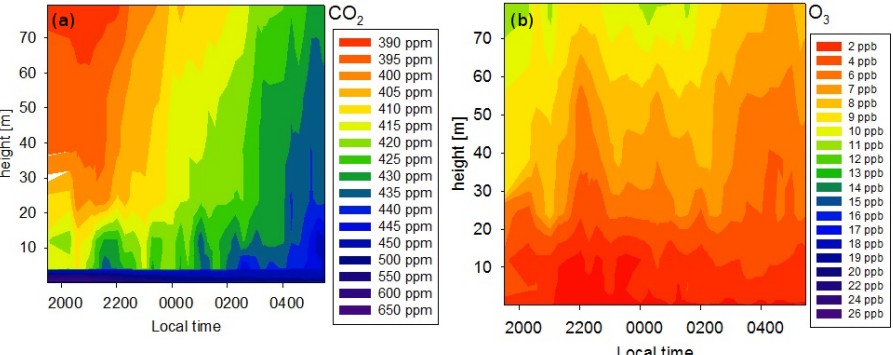

**Figure 3.** Concentrations of $CO_2$ (a) and $O_3$ (b) as a function of time and height for the turbulent night.

In the intermittent night, on the other hand, large increases and decreases of all scalars occur in small periods of time at all
levels, except at 80 m. As $CO_2$ has a clear source at the ground and $O_3$ has clear sink at the ground one can identified from the
profiles if air is coming from aloft or from below (Fig. 4). Air from above is rich in $O_3$ and lower in $CO_2$, whereas air from
below is rich in $CO_2$ but depleted in $O_3$. From this perspective, in the first event air is mixed down from aloft, while in the
5    second event air is mixed both upward and downward from the canopy top. In the third event, air is first mixed down and finally
there is a burst of air going upwards from the canopy. At 80 m, temperature (Fig. 2b) and $CO_2$ (Fig. 2d) show much smaller
fluctuations than at the other levels. This is further evidence that the stable boundary layer (SBL) thickness is shallower during
the intermittent night, such that the canopy exchange fluxes do not affect the state of the atmosphere at 80 m. This fact contrasts
strongly with the steady trends of both scalars at 80 m during the turbulent night (Fig. 1b and Fig. 1d), which indicates that in
10   this case, this level is fully coupled through turbulence to the canopy top. While in the turbulent nighttime, scalar fluxes did
not vary substantially throughout the period (Fig. 5a, c, e, g), the most intense turbulent fluxes of sensible heat (Fig. 5b), $CO_2$
(Fig. 5d), $O_3$ (Fig. 5f), and latent heat (Fig. 5h) during the intermittent night occurred during these coupling periods (Table 2).

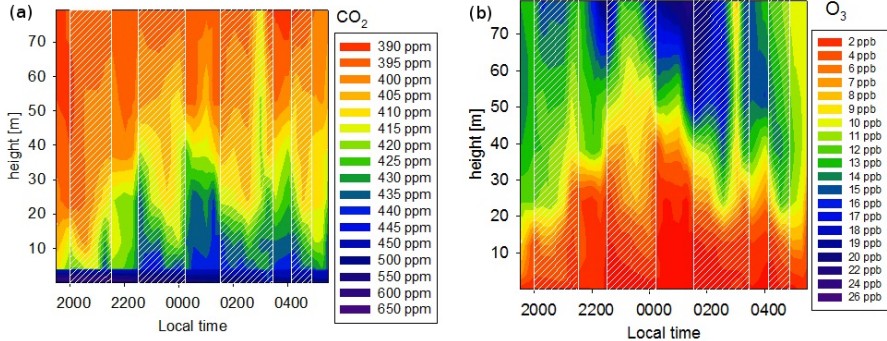

**Figure 4.** Same as in Fig. 3, but for the intermittent night.





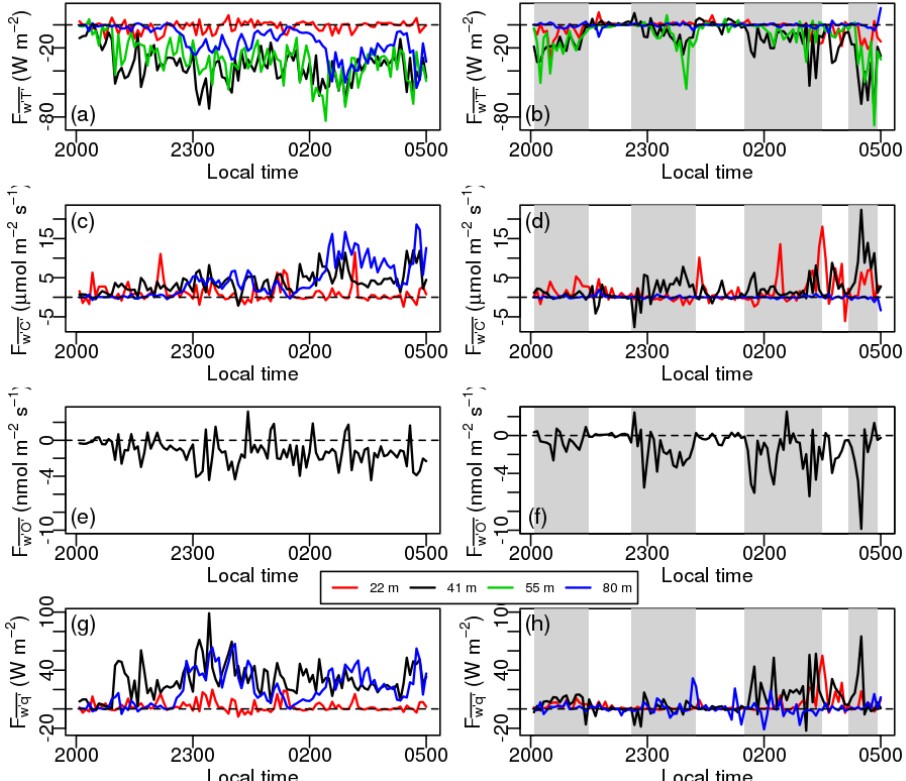

**Figure 5.** Fluxes of sensible heat (a and b), $CO_2$ (c and d), $O_3$ (e and f), and latent heat (g and h) for the turbulent night (left panels) and for the intermittent night (right panels). Shaded areas indicate intermittent turbulence bursts as shown in Fig. 2.

Previous studies have reported that non-turbulent modes of the flow only become relevant when turbulence is weak (Acevedo et al., 2014), a likely consequence of the diffusive nature of turbulence destroying the non-turbulent temporal and spatial variability of the atmospheric variables. This relationship between the turbulent and non-turbulent modes of the flow is illustrated by the TKE spectrum during both nights (Fig. 6). In the turbulent case, most of the energy is associated with turbulence, so

5   that the most energetic time scale is near 10 s at all levels, except for 80 m (Fig. 6a). At this level, the longest time scales are the most energetic, but the 10-s turbulent maximum and a cospectral gap (near 100 s) are still evident. In contrast, in the intermittent night of November 14, at all levels most energy prevails at the longest timescales provided by the decomposition method. This energy is associated with the low-frequency fluctuations responsible for the variability of the wind direction visible in Fig. 2a and Fig. 2c. These spectra confirm that when fully turbulent conditions prevail, the energy of the non-turbulent, low frequency modes of the flow is reduced considerably. The cospectra of the fluxes of sensible heat (Fig. 7a, d, f, i), $CO_2$

10  (Fig. 7b, g, j), $O_3$ (Fig. 7e), and latent heat (Fig. 7c, h, k) confirm the enhanced turbulent exchange of all quantities in the turbulent night compared to the intermittent one. They also show that, consistently to what occurs with TKE, the non-turbulent exchange of these scalars is enhanced in the intermittent case. In particular, a significant low-frequency flux of $CO_2$ occurs





at 22 m in the intermittent night, such that the total flux at this level is larger during the intermittent night (4.0 $\mu$mol m$^2$ s$^{-1}$, Table 3) than during the turbulent one (1.1 $\mu$mol m$^2$ s$^{-1}$), when all scales of the motion that are captured by the decomposition window are considered. This is in line with the idea that non-turbulent motions better penetrate the canopy. The same occurs for latent heat, which shows a mean flux of 8.8 W m$^{-2}$ in the intermittent night and of 2.8 W m$^{-2}$ in the turbulent night, when

all scales are considered. Even for 5-min fluxes, the larger fluxes occur still in the intermittent night (1.7 $\mu$mol m$^2$ s$^{-1}$ versus 1.0 $\mu$mol m$^2$ s$^{-1}$ in the turbulent night for $CO_2$, and 3.8 W m$^{-2}$ versus 2.5 W m$^{-2}$ for latent heat), but the differences are smaller. These numbers show that the low-frequency contributions dominate the exchange of $CO_2$ and moisture in the interior of the forest in the intermittent night. Santos et al. (2016) found that the time scales of horizontal turbulent velocity fluctuations within an Amazonian rain forest canopy (at a different site) approach those of the non-turbulent maximum above the forest.

They also found that the dominant time scales of the vertical velocity fluctuations and sensible heat flux within the canopy are shifted towards larger values than those above it. Our results support these findings, adding the information that the non-turbulent contribution may dominate the exchange of $CO_2$ and humidity from the interior of the canopy in very stable nights as well. It is likely that the same process affects other scalars, such as $O_3$, whose concentrations are perturbed by intermittent events as shown in Fig. 4b.

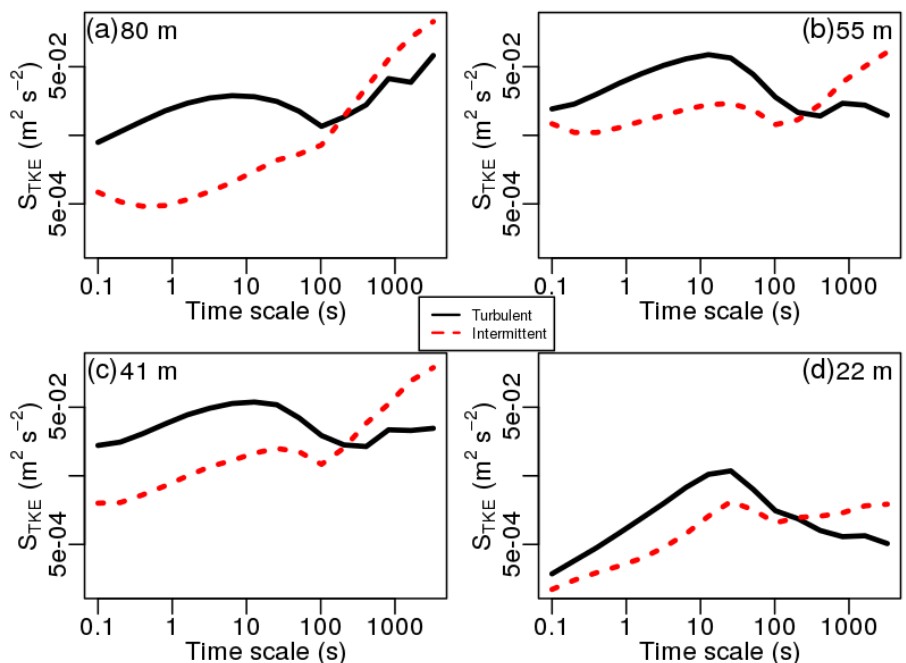

**Figure 6.** Average TKE spectra for the turbulent night (black solid lines) and the intermittent night (red dashed lines) for all levels, as indicated in each panel.

Another interesting contrast between the turbulent and intermittent nights can be seen for the sensible heat (Fig. 7a), $CO_2$ (Fig. 7b), and latent heat (Fig. 7c) cospectra at 80 m, which show an almost total absence of turbulent fluxes (time scale smaller





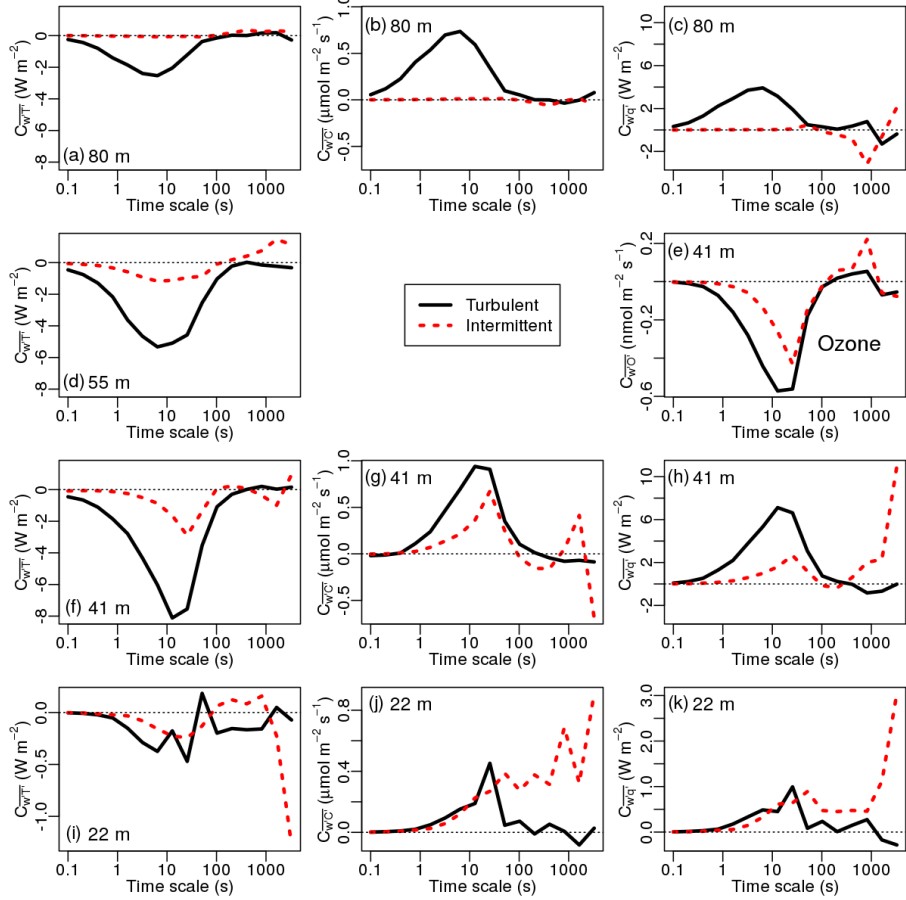

**Figure 7.** Average cospectra of sensible heat (left panels), $CO_2$ (central panels), $O_3$ (e), and latent heat fluxes (c, h and k) for the turbulent night (black solid lines) and the intermittent night (red dashed lines) for all levels, as indicated in each panel.

than 100 s) of these quantities at this height in the intermittent case. This result explains the reduced fluctuations of temperature (Fig. 2b) and $CO_2$ (Fig. 2d) at 80 m in the intermittent night, adding evidence to the suggestion that the SBL may be rather shallow in that night, possibly such that the 80-m level is near its top. The large fluctuations of water vapor at 80 m (Fig. 2f) may be related to the enhanced low frequency flux of this quantity during this night (Fig. 7c).

Sun et al. (2012) found two regimes of nocturnal turbulence, distinguished by the turbulent kinetic energy (TKE) dependence on the mean wind speed. The fully turbulent regime, typically associated with weakly stable conditions, happens for mean wind speeds larger than a height dependent threshold and is characterized by TKE that steadily increases with wind speed. The other regime, associated with very stable conditions, has reduced turbulence intensities, which are very weakly dependent on the mean wind speed. Dias-Júnior et al. (2017) observed the two regimes above the forest at a site in the southwestern Amazon, finding that each is associated with an independent lognormal frequency distribution of quantities such as the turbulence dissipation rate. For the turbulent night of 15 November 2015 (Fig. 8, crosses), the levels of 41 and 55 m remained in the large





wind speed regime for the whole period, while the two different regimes could be observed only at the 80-m level. On the intermittent night, on the other hand (Fig. 8, triangles), both regimes could be observed at all levels. Moreover, the connection intervals, given by shaded areas in Fig. 2, are generally in the large wind speed regime both at 41 m and 55 m (Fig. 8, filled triangles), while the decoupled periods are in most cases in the weak wind regime (Fig. 8, open triangles). This is an important

result, because it indicates that the intermittent bursts of turbulence observed above the canopy are intense enough to cause a regime transition. It means that, during these events, there is likely full vertical coupling over the vertical extent of the SBL (which is, at this time, shallower than 80 m). Therefore, scalars that are emitted from the canopy may be able to escape to higher levels in the boundary layer, as suggested by the episodic mixing of $CO_2$ and $O_3$ above 70 m shown in Fig. 4.

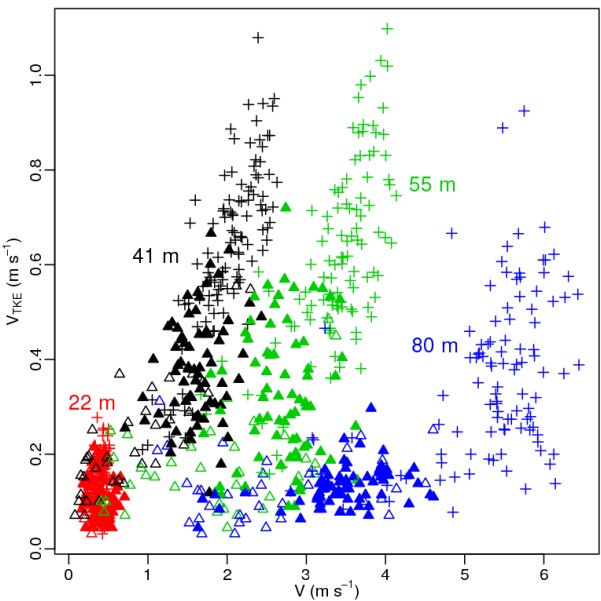

**Figure 8.** Turbulent velocity scale ($V_{TKE}$) dependence on mean wind speed ($V$) for the turbulent night (crosses), shaded areas in Fig. 2 (filled triangles) and non-shaded areas in Fig. 2 (open triangles).

## 4   Mean spectra and cospectra

Figure 9 shows the spectra and cospectra of the turbulent fluctuations and fluxes averaged over the entire period. The distinction between turbulent and non-turbulent fluctuations across the vertical is evident in the averaged TKE spectra (Fig. 9a). On average, a time scale around 100 s separates the two types of fluctuations. This can be seen by the fact that $S_{TKE}$ decreases with height above the canopy, following what typically happens to turbulent fluctuations in the stable boundary layer (Kruijt et al., 2000; Santos et al., 2016) only for time scales smaller than 100 s. For longer time scales, $S_{TKE}$ increases with height.

This result has been previously shown by Andreae et al. (2015) at the same tower, but for a different period. The average



cospectra of all scalars show the largest turbulent fluxes at 41 m, with a sharp maximum at a timescale around 30 s. Within the canopy, systematic positive fluxes of $CO_2$ and latent heat happen at all time scales. The magnitude increases with increasing time scale. Therefore, this is an indication that the $CO_2$ and humidity exchanges in the interior of the forest may be, to a large extent, caused by non-turbulent motion. This process will be further investigated later in this paper, when the scale dependence

5   of the fluxes is compared to the stability within and above the canopy. The average sensible heat cospectrum at 22 m is negative for time scales smaller than 50 s and positive for larger time scales. Santos et al. (2016) showed that sensible heat fluxes near the forest floor are positive for all time scales, while they are negative for all time scales at the canopy top. At intermediate heights, they tend to be like those shown in Fig. 9c. These authors also showed that the height within the canopy where the total sensible heat flux switches sign from upward at lower levels to downward at higher levels is stability dependent, increasing as

10   it becomes more stable.

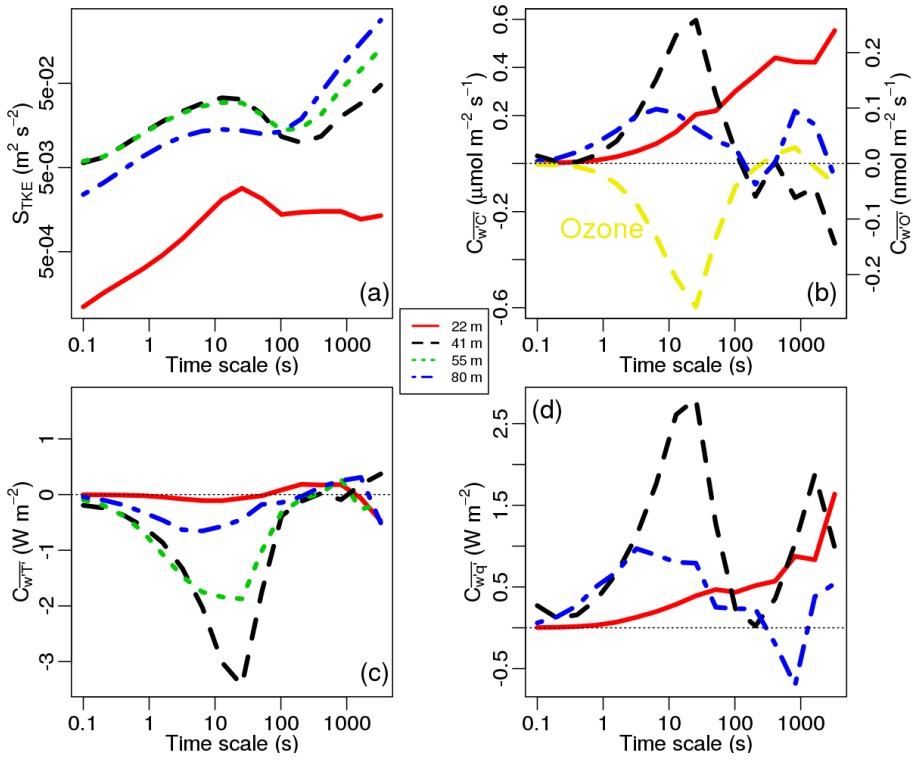

**Figure 9.** Average spectra of $TKE$ (a) and cospectra of $CO_2$ and $O_3$ (b), sensible heat (c), and latent heat (d) fluxes for the entire dataset.

## 5   Dependence on stability

The comparison of the fluxes determined with 5-min and 109-min time windows and their stability dependence provides interesting information on the scalar exchange processes within and above the canopy. Sensible heat flux at 22 m (Fig. 10i)





switches sign at intermediate stability for both 5-min and 109-min averaging periods. This is in agreement with the result obtained by Santos et al. (2016), that the height where the sensible heat flux switches sign is stability dependent. In the present case, the critical value at which the 5-min sensible heat flux at 22 m switches sign is 0.25. At the same height, $CO_2$ (Fig. 10j) and latent heat (Fig. 10k) fluxes are similar with both averaging times at near-neutral conditions, but become appreciably larger

with 109-min windows than with 5-min windows as conditions become more stable. This result confirms the idea that the $CO_2$ and humidity exchange within the canopy at very stable conditions are dominated by processes with long time scales. At 41 m (Fig. 10e, f, g, h), 55 m (Fig. 10d), and 80 m (Fig. 10a, b, c) the 109-min fluxes of scalars are more erratic, sometimes with no clear dependence on $Ri_{can}$. The 5-min fluxes, on the other hand show a tendency to decrease in magnitude with stability at all heights above the canopy. The 5-min TKE decreases with stability at all levels (Fig. 11), but at 80 m it becomes near

zero for $Ri_{can} > 0.2$. This is the same range of stability for which the 5-min fluxes of heat and $CO_2$ are virtually suppressed, indicating that in these very stable conditions the stable boundary layer thickness may be close to 80 m or shallower. When the same quantities are compared to $Ri_{top}$ (Fig. 12), the most significant difference is that the 109-min fluxes at 41 m (Fig. 12e, f, g, h, black lines) and 55 m (Fig. 12d, black line) show a more systematic dependence on stability. This result indicates that the low-frequency exchange at the canopy top and above is controlled by the stability over a large distance above the forest.

The 0.25-$Ri_{can}$ threshold, over which the 22-m turbulent sensible heat flux switches sign, is used to classify each series as weakly stable ($Ri_{can} \leq 0.25$) or very stable ($Ri_{can} > 0.25$). The average spectra for each of these classes are shown in Fig. 13. At all levels, the low-frequency TKE is almost independent of stability, as for time scales larger than 100 s the TKE spectra of the very stable series approach those of the weakly stable ones. For smaller time scales, on the other hand, a significant distinction prevails, with TKE being many times larger under weakly stable conditions than during very stable conditions. The

turbulent portions of the scalar fluxes above the canopy (41 m and above) respond accordingly, always being much larger in magnitude in the weakly stable cases (Fig. 14).

At 22 m, the criterion for classifying the time series ensures that the sensible heat flux at small time scales must be negative under the weakly stable conditions and positive under the very stable ones. This is shown in Fig. 14i, but it is remarkable that even in the weakly stable cases, there is a range of time scales for which the sensible heat flux within the canopy is upward.

This range is also observed (however broader) in the very stable class, although in this case the negative maximum associated with turbulent exchange is absent. This result shows that low-frequency exchange of heat within the canopy is consistently upward, regardless of stability. Therefore, the main control exerted by stability regards the depth within the canopy where the downward turbulent heat flux penetrates (Santos et al., 2016). The total 22-m fluxes of $CO_2$ (Fig. 14j) are larger in the very stable class than in the weakly stable one, which is mainly caused by the contribution of time scales larger than 100 s,

corroborating the results from the case studies (Sect. 3). For latent heat (Fig. 14k), the total flux is slightly larger in the weakly stable cases, and this result is mainly caused by the contributions of fluxes with time scales smaller than 100 s.




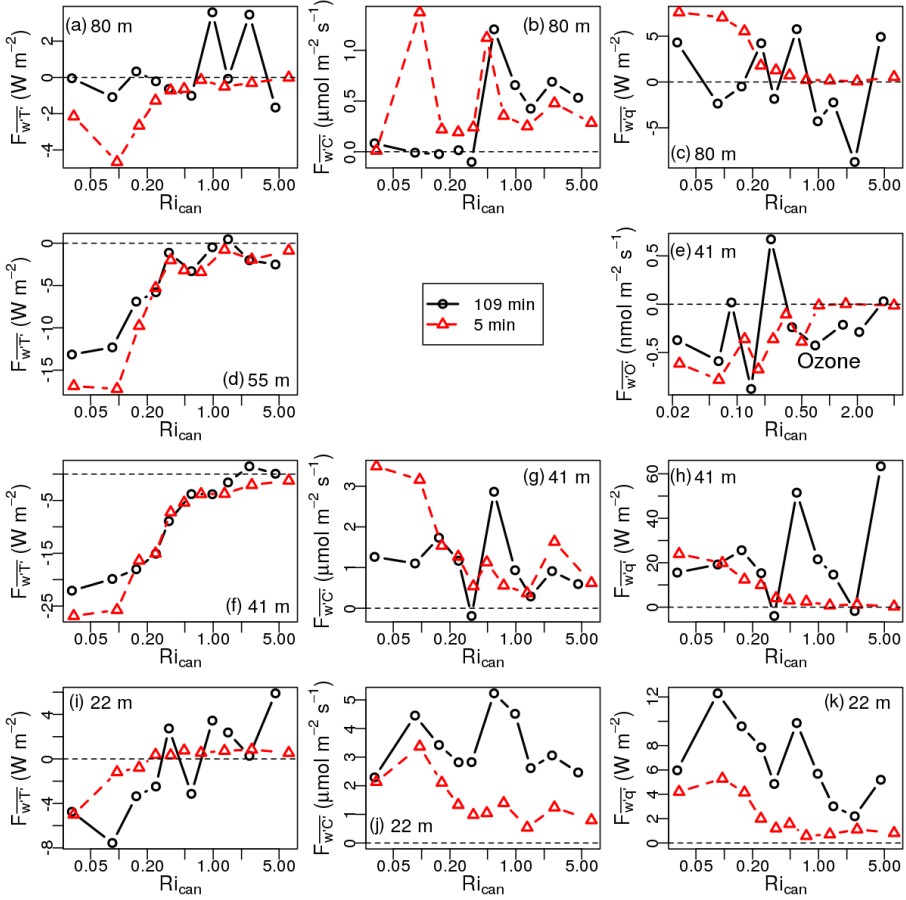

**Figure 10.** The dependence of sensible heat (left panels), $CO_2$ (central panels), $O_3$ (e), and latent heat (c, h and k) fluxes on canopy Richardson number ($Ri_{can}$) for all levels, as indicated in each panel. Fluxes have been determined with 5-min (red lines, triangles) and 109-min (black lines, circles) time windows.

## 6 Conclusions

The main novelty of the present study has been a detailed analysis of different scalar fluxes and their time scales within and above a rain forest canopy at night. The data was collected at the Amazon Tall Tower Observatory (ATTO) and included fluxes of $CO_2$, $O_3$, latent and sensible heat. The most relevant findings include:

5      – Within the canopy, fluxes of $CO_2$ and latent heat are dominated by processes with long time scales. Given that such low-frequency exchange tends to be enhanced in very stable conditions, the total scalar flux within the canopy may be larger in very stable nights than in weakly stable ones;



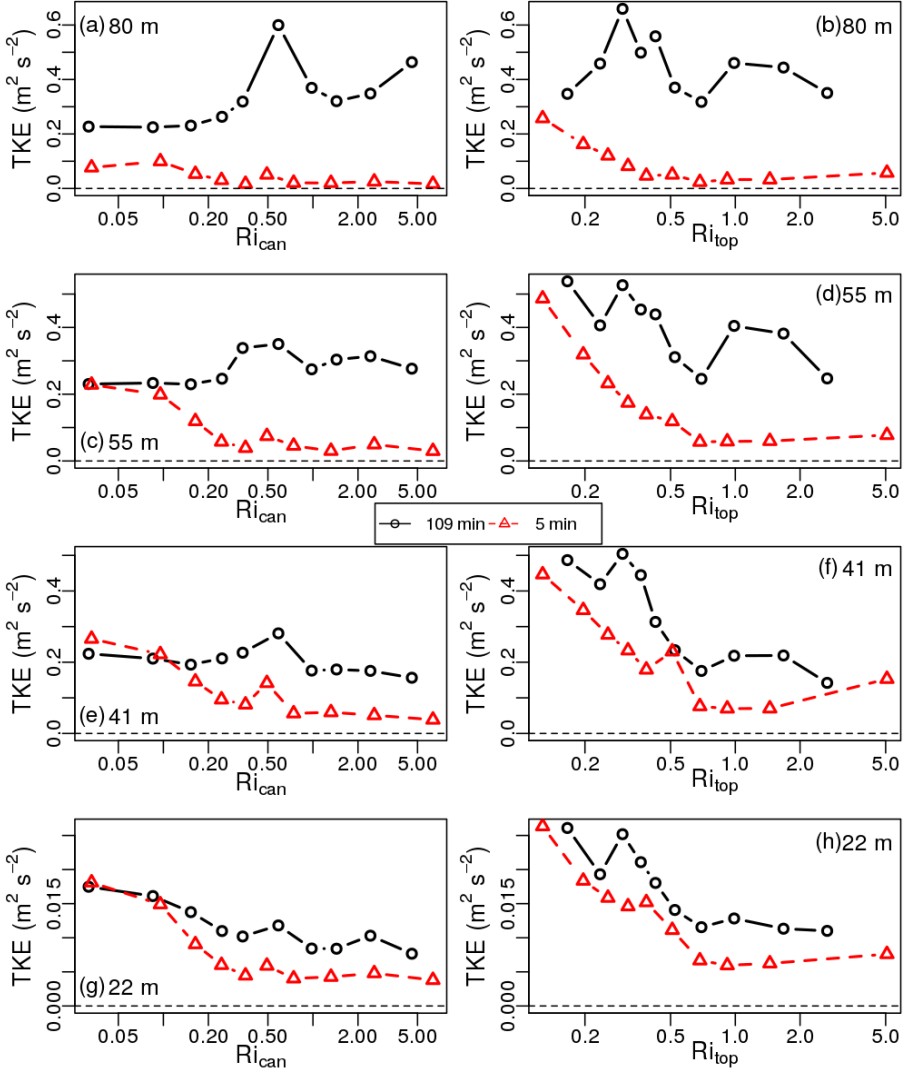

**Figure 11.** The dependence of TKE on canopy Richardson number ($Ri_{can}$, left panels) and top Richardson number ($Ri_{top}$, right panels) for all levels, as indicated in each panel, using 5-min (red lines, triangles) and 109-min (black lines, circles) time windows.

- In very stable nights, turbulent fluxes are effectively suppressed at 80 m, indicating that a very shallow stable boundary layer (SBL) may exist in these situations;

- Intermittent turbulence may produce very large fluxes and affect concentrations of $CO_2$ and $O_3$ from near the SBL top down to the middle of the canopy.




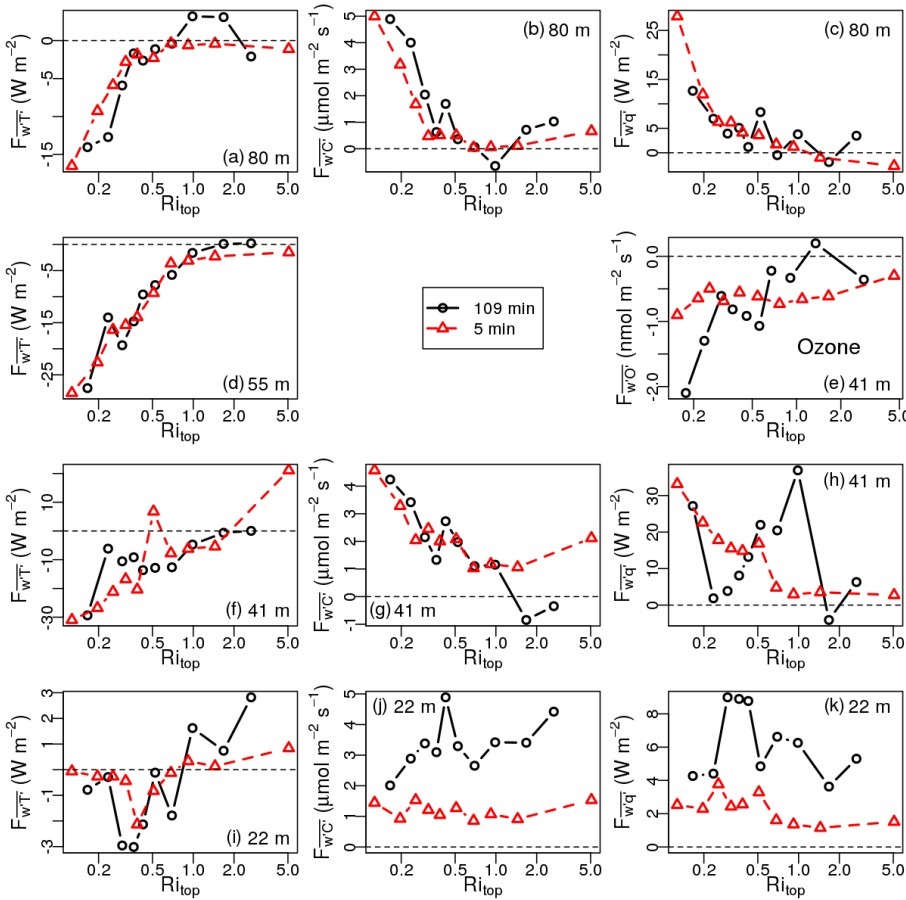

**Figure 12.** Same as in Fig. 10, but for dependence on top Richardson number ($Ri_{top}$).

Although low-frequency contributions to the fluxes are enhanced during very stable nights, their inclusion into the scalar budgets is not enough to bring the nocturnal fluxes in the very stable nights close to those observed during fully turbulent conditions. Processes such as drainage flows or local storage may account for the differences.

The majority of the flux just above the canopy (41-m level, in this case) happens through turbulent exchange. Although 5  no relevant systematic low-frequency contribution to the total fluxes has been found at 41 m for any scalar analyzed (Fig. 9), this result only holds for the average spectra, with appreciable variability among cases, especially in the most stable cases. Campos et al. (2009) found the low-frequency contribution of $CO_2$ fluxes above a similar Amazonian canopy to be seasonally dependent, a result that could not be examined with the present data set.

A fully instrumented 320-m tower is expected to start operating continually in the near future at the ATTO site. It will allow 10  addressing questions such as the seasonality of the exchange at different scales, as well as the thickness of the SBL and the nature of the scalar exchange within and above the canopy in much more detail. In this sense, the results of the present study will provide important guidelines for the future investigations.





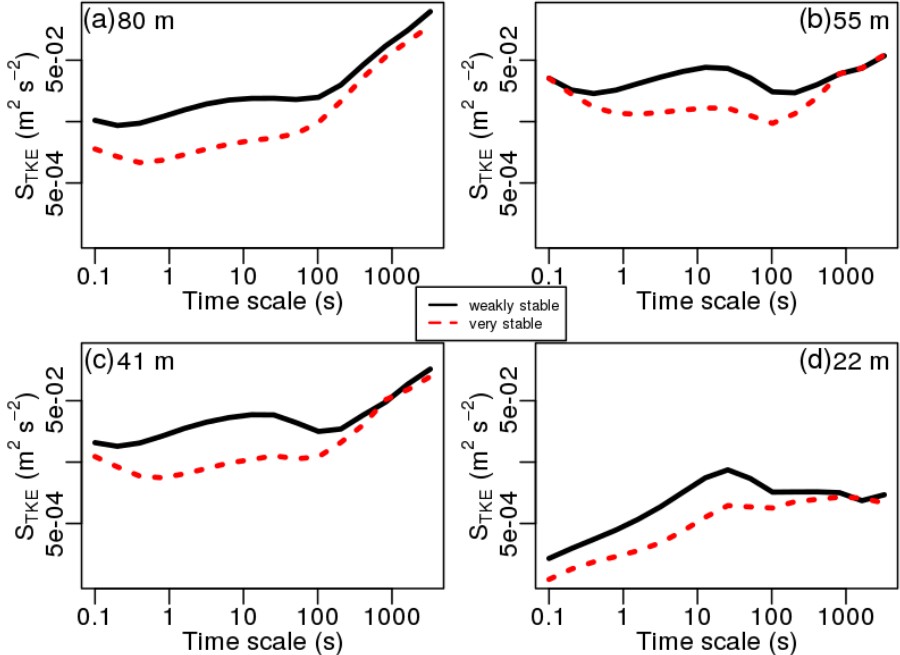

**Figure 13.** Average TKE spectra for the weakly stable (black solid lines) and the very stable (red dashed lines) cases for all levels, as indicated in each panel.

*Data availability.* All data used in this study are kept in the ATTO Databases at Instituto de Pesquisas da Amazônia and Max Planck Institut Für Chemie. The overall project description can be found at http://www.mpic.de/en/research/collaborative-projects/atto.html. Data access can be requested from the coauthors responsible for maintaining the dataset: Matthias Sörgel (m.soergel@mpic.de) and Alessandro Araújo (alessandro.araujo@gmail.com).

5   *Acknowledgements.* For the operation of the ATTO site, we acknowledge the support by the German Federal Ministry of Education and Re-
search (BMBF contract 01LB1001A) and the Brazilian Ministério da Ciência, Tecnologia e Inovação (MCTI/FINEP contract 01.11.01248.00)
as well as the Amazon State University (UEA), FAPEAM, LBA/INPA and SDS/CEUC/RDS-Uatumã. This work was in particular supported
by the Max Planck Society (MPG), the Instituto Nacional de Pesquisas da Amazônia (INPA), and the Conselho Nacional de Desenvolvimento
Científico e Tecnológico (CNPq). We would like to thank Rodrigo Souza from the Universidade do Estado do Amazonas (UEA) for lending
10  us the slow-response $O_3$ analyzer.





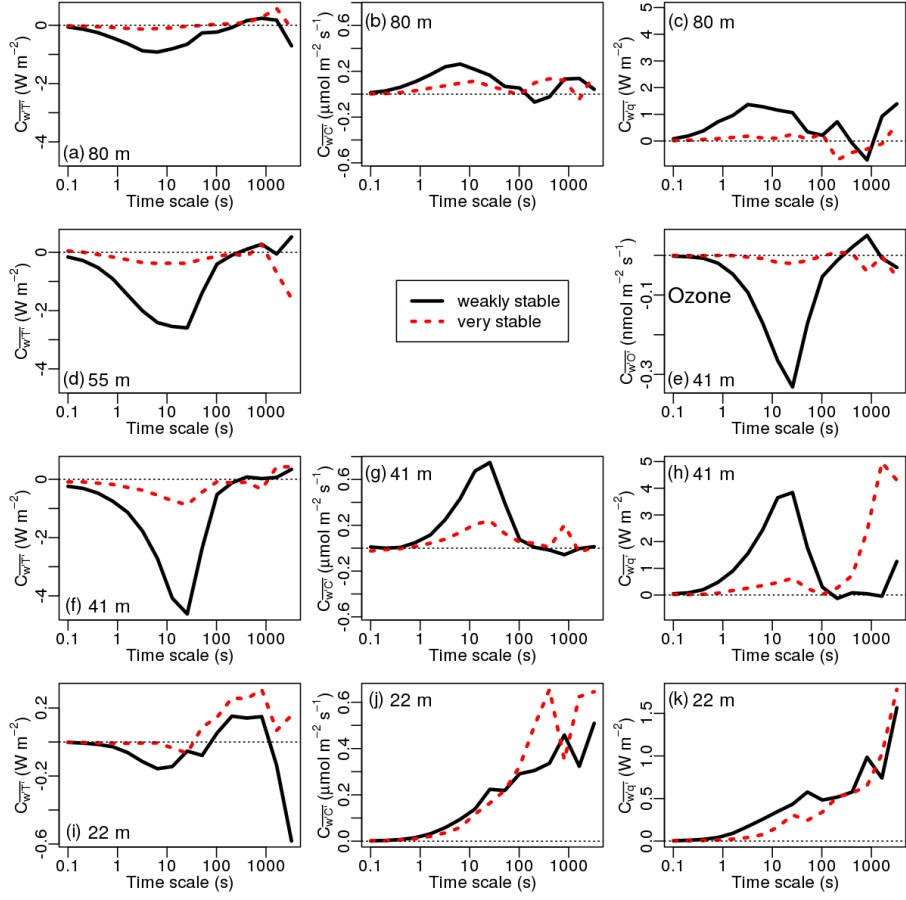

**Figure 14.** Average cospectra of sensible heat (left panels), $CO_2$ (central panels), $O_3$ (e), and latent heat fluxes (c, h and k) for the weakly stable (black solid lines) and the very stable (red dashed lines) cases for all levels, as indicated in each panel.

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





**Table 1.** 5-min turbulence statistics averaged for each night analyzed in Sect. 3.

| | 14 November 2015 Intermittent night | | | 15 November 2015 Turbulent night | | |
|---|---|---|---|---|---|---|
| level | $\sigma_\mathrm{w}$ | $u_*$ | TKE | $\sigma_\mathrm{w}$ | $u_*$ | TKE |
| (m) | $(\mathrm{m\,s^{-1}})$ | $(\mathrm{m\,s^{-1}})$ | $(\mathrm{m^2\,s^{-2}})$ | $(\mathrm{m\,s^{-1}})$ | $(\mathrm{m\,s^{-1}})$ | $(\mathrm{m^2\,s^{-2}})$ |
| 22 | 0.07 | 0.04 | 0.01 | 0.11 | 0.07 | 0.03 |
| 41 | 0.19 | 0.14 | 0.13 | 0.39 | 0.30 | 0.44 |
| 55 | 0.15 | 0.10 | 0.10 | 0.37 | 0.27 | 0.41 |
| 80 | 0.06 | 0.04 | 0.02 | 0.18 | 0.13 | 0.16 |





**Table 2.** 5-min TKE and fluxes averaged for shaded and non-shaded areas in the intermittent night (see Fig. 2).

| | level | $F_C$ | $F_H$ | $F_q$ | TKE | $F_O$ |
|---|---|---|---|---|---|---|
| | (m) | $\mu$mol m$^2$ s$^{-1}$ | W m$^{-2}$ | W m$^{-2}$ | (m$^2$ s$^{-2}$) | nmol m$^2$ s$^{-1}$ |
| **Shaded** | 22 | 1.6 | -2.9 | 3.7 | 0.01 | |
| | 41 | 2.6 | -15.9 | 9.6 | 0.16 | -1.5 |
| | 55 | - | -12.5 | - | 0.13 | |
| | 80 | 0.0 | -0.4 | 0.9 | 0.02 | |
| **Non-shaded** | 22 | 1.6 | -0.4 | 3.7 | 0.01 | |
| | 41 | 1.0 | -3.2 | 2.9 | 0.06 | -0.4 |
| | 55 | - | -2.4 | - | 0.03 | |
| | 80 | 0.0 | -0.1 | 0.4 | 0.02 | |

14 November 2015 – Intermittent night





**Table 3.** TKE and fluxes averaged for each night analyzed in Sect. 3 using a time window of 5 and 109 min.

| | level | $F_C$ | $F_H$ | $F_q$ | TKE | $F_O$ | $F_C$ | $F_H$ | $F_q$ | TKE | $F_O$ |
|---|---|---|---|---|---|---|---|---|---|---|---|
| | | | 14 November 2015 | | | | | | 15 November 2015 | | |
| | | | Intermittent night | | | | | | Turbulent night | | |
| | (m) | µmol m$^2$ s$^{-1}$ | W m$^{-2}$ | W m$^{-2}$ | (m$^2$ s$^{-2}$) | nmol m$^2$ s$^{-1}$ | µmol m$^2$ s$^{-1}$ | W m$^{-2}$ | W m$^{-2}$ | (m$^2$ s$^{-2}$) | nmol m$^2$ s$^{-1}$ |
| **109 min** | 22 | 4.0 | -2.0 | 8.8 | 0.02 | | 1.1 | -2.1 | 2.8 | 0.03 | |
| | 41 | 1.2 | -7.9 | 22.9 | 0.29 | -0.9 | 3.5 | -2.1 | 29.7 | 0.49 | -2.3 |
| | 55 | - | -2.4 | - | 0.48 | | - | -37.4 | - | 0.49 | |
| | 80 | -0.1 | 1.0 | -2.4 | 0.47 | | 3.9 | -13.4 | 20.4 | 0.29 | |
| **5 min** | 22 | 1.6 | -2.1 | 3.7 | 0.01 | | 1.0 | -1.3 | 2.6 | 0.03 | |
| | 41 | 2.1 | -11.9 | 7.5 | 0.13 | -1.2 | 3.8 | -35.6 | 29.4 | 0.44 | -1.3 |
| | 55 | - | -9.3 | - | 0.10 | | - | -30.2 | - | 0.41 | |
| | 80 | 0.0 | -0.3 | 0.7 | 0.02 | | 4.3 | -14.1 | 20.8 | 0.16 | |