# Peer review of "Nighttime wind and scalar variability within and above an Amazonian canopy"

_Atmospheric Chemistry and Physics, 2017_

## Referee Comment (RC1) · Anonymous Referee #1 · 10 Aug 2017

This paper analyzes wind and scalar turbulence measurements during several nights at the ATTO project site. A great deal of the analysis is about comparisons between two specific nights. One of these nights is classified as being "fully turbulent", and the other as displaying "intermittent turbulence". Most of the analyses are made using multiresolution decomposition.

The results are interesting and should be useful to understand nighttime scalar exchanges between the forest and the atmosphere. However, the text needs a significant reorganization, as the comparisons between the two nights and the several heights proceed in a rather disorderly way. In this regard, I recommend that all discussions

start with the turbulent night and proceed whenever possible level by level; that the same be done for the intermittent night; and that, finally, comparisons between the two nights are made. Most of the time, this should be done in different paragraphs. This will enhance readability significantly.

Moreover ("major issues"),

1. The text is ambiguous about the role of the low frequencies' contribution to the above-canopy fluxes.

2. Gradients of temperature and velocity are being used in the Richardson numbers, but no mention to the systematic errors in the measurements between the levels is made. This should be addressed.

3. Turbulent bursts and activity are not quantitavely defined.

4. The discussion starting on p. 11, l. 5, on the turbulent regimes seems to be a re-packaging of results already presented in the manuscript. It does not seem to bring any new information.

5. Clear indication must be given when only 2 nights are being compared and when all data are being used.

6. The effect of averaging per frequency without taking stability into account should be investigated.

Please also note the supplement to this comment:
https://www.atmos-chem-phys-discuss.net/acp-2017-638/acp-2017-638-RC1-supplement.pdf

**Supplement:**

**Comments on "Turbulent and non-turbulent exchange of scalars between the forest and the atmosphere at night in Amazonia", by Oliveira et al.**

August 10, 2017

**General remarks**

This paper analyzes wind and scalar turbulence measurements during several nights at the ATTO project site. A great deal of the analysis is about comparisons between two specific nights. One of these nights is classified as being "fully turbulent", and the other as displaying "intermittent turbulence". Most of the analyses are made using multiresolution decomposition.

The results are interesting and should be useful to understand nighttime scalar exchanges between the forest and the atmosphere. However, the text needs a significant reorganization, as the comparisons between the two nights and the several heights proceed in a rather disorderly way. In this regard, I recommend that all discussions start with the turbulent night and proceed whenever possible level by level; that the same be done for the intermittent night; and that, finally, comparisons between the two nights are made. Most of the time, this should be done in different paragraphs. This will enhance readability significantly.

Moreover ("major issues"),

1. The text is ambiguous about the role of the low frequencies' contribution to the above-canopy fluxes.

2. Gradients of temperature and velocity are being used in the Richardson numbers, but no mention to the systematic errors in the measurements between the levels is made. This should be addressed.

3. Turbulent bursts and activity are not quantavely defined.

4. The discussion starting on p. 11, l. 5, on the turbulent regimes seems to be a re-packaging of results already presented in the manuscript. It does not seem to bring any new information.

5. Clear indication must be given when only 2 nights are being compared and when all data are being used.

6. The effect of averaging per frequency without taking stability into account should be investigated.

**Recommendations**

In view of the above, I recommend a major review of the current manuscript.

**Major issues**

1. On page 2, l. 21–23, the authors say:

   > Equivalent analyses focusing on scalar flux cospectra have not been presented as often. Sakai et al. (2001) and Finnigan et al. (2003) used cospectral similarity to conclude that low-frequency contribution could account for missing energy and CO2 fluxes in their respective budgets, but neither study addressed how the cospectra varied across the canopy.

   later, on p. 2, l. 30–33, they say:

   > This result indicates that the exchange of scalars between the canopy and the atmosphere at night may occur at longer time scales than those traditionally used in the eddy covariance approach.

   and again, on p. 10, l. 10–15:

   > Our results support these findings, adding the information that the **non-turbulent contribution may dominate the exchange of CO2 and humidity from the interior of the canopy in very stable nights as well**. It is likely that the same process affects other scalars, such as O3 , whose concentrations are perturbed by intermittent events as shown in Fig. 4b.

   However, in the conclusions, they find that low-frequency components are important within the canopy, but that, above the canopy, it is the "turbulent scales" that contribute most of the flux. There seems to be a contradiction between the Introduction (and other parts of the manuscript) and the Conclusions. The introduction should not lead the reader to believe in a situation that will not be supported by the analysis.

2. "Bulk" Richardson numbers are used, but these are sensitive to velocity and, most of all, temperature systematic errors between the sensors. Because several analyses are dependent on these Richardson numbers, their reliability must be assessed quantitatively. Have the sensors been intercompared?

   In the worst case (no intercomparison, no calibration), a thorough sensitivity analysis must be made of the effects of the temperature (and wind) systematic errors on those Richardson numbers and in the analyses involving them. The reported accuracies for the sensors (assuming optimistically that they did not drift) can be used as a basis for this. The uncertainty introduced by those errors results should then be displayed graphically in all analyses regarding the Richardson number.

3. Turbulent bursts: the criterion for identifying the turbulent bursts and defining the shaded regions in Fig. 2 should be made clear (quantitavely).

4. Text starting on p. 11, l. 5, says

> Sun et al. (2012) found two regimes of nocturnal turbulence, distinguished by the turbulent kinetic energy (TKE) dependence on the mean wind speed. The fully turbulent regime, typically associated with weakly stable conditions, happens for mean wind speeds larger than a height dependent threshold and is characterized by TKE that steadily increases with wind speed. The other regime, associated with very stable conditions, has reduced turbulence intensities, which are very weakly dependent on the mean wind speed. Dias-Júnior et al. (2017) observed the two regimes above the forest at a site in the southwestern Amazon, finding that each is associated with an independent lognormal frequency distribution of quantities such as the turbulence dissipation rate. For the turbulent night of 15 November 2015 (Fig. 8, crosses), the levels of 41 and 55 m remained in the large wind speed regime for the whole period, while the two different regimes could be observed only at the 80-m level. On the intermittent night, on the other hand (Fig. 8, triangles), both regimes could be observed at all levels. Moreover, the connection intervals, given by shaded areas in Fig. 2, are generally in the large wind speed regime both at 41 m and 55 m (Fig. 8, filled triangles), while the decoupled periods are in most cases in the weak wind regime (Fig. 8, open triangles). **This is an important result, because it indicates that the intermittent bursts of turbulence observed above the canopy are intense enough to cause a regime transition. It means that, during these events, there is likely full vertical coupling over the vertical extent of the SBL (which is, at this time, shallower than 80 m). Therefore, scalars that are emitted from the canopy may be able to escape to higher levels in the boundary layer, as suggested by the episodic mixing of $CO_2$ and $O_3$ above 70 m shown in Fig. 4.**

(my emphasis). But high turbulent fluxes above the canopy during the the bursts of turbulent activity are already clearly displayed in Fig 5. The "full coupling" is none other than the relatively high (absolute) values of the fluxes themselves. Given that the fluxes are there, the scalars have already "escaped" the canopy. Therefore, the reasoning in the bold-face text above seems to be rather circular, and nothing new seems to arise from this discussion. Moreover, if the criterion for identifying the bursts was TKE (as I suspect), then it is inevitable that this will be reflected in higher TKE values in Fig. 8. It appears to me that the definition of the bursts and the regime classification in Fig. 8 are one and the same, and that there is nothing to be added here. I strongly suggest deleting this whole passage.

5. Sections 4 and 5 seem to use all the data from the 15 usable nights. Because the previous section focused strongly on the comparison of the nights of Nov 14 and 15, I had a hard time (at first reading) realizing this. I suggest that both the title and the introduction of each of these sections reinforces the information that, now, data from all 15 nights are being analyzed.

6. (p. 12, l. 10): "Figure 9 shows the spectra and cospectra of the turbulent fluctuations

and fluxes averaged over the entire period".

Particularly in stable conditions, there is a strong shift of the spectra towards the higher frequencies with increasing stability (Kaimal, 1973). There is no equation describing how the spectra were "averaged", but there should be. The simplest approach (which I suspect is being used here) is to average per frequency. But then, because frequency depends on stability, different stabilities and their spectral densities are being averaged together. The consequences are far from clear to me, and this procedure should not be done without careful justification.

Remember, if $y = f(x)$ and $f$ is nonlinear, then $\overline{y} \neq f(\overline{x})$ in general. It is not clear how the fluxes reported in Sect. 5 were calculated. Are they bin averages? Do they come from the integration of the *mean* spectra? If $F_{wa}^{(i)}$ is the flux from the $i^{\text{th}}$ cospectrum, and if $F_{wa,\text{mean}}$ is the flux from the *mean* cospectrum (as depicted in Fig. 9), how do $(1/n) \sum_{i=1}^{n} F_{wa}^{(i)}$ and $F_{wa,\text{mean}}$ compare? In this sense, how valuable and correct are the conclusions derived from Fig. 9?

**Specific comments**

p. 4, l. 15–16:  "Since the different levels of flow structures are analyzed simultaneously, only the data when all levels were available was used."

This should be: "…Since the different levels of flow structures are analyzed simultaneously, only the data when all levels were available **were** used".

p. 4, l. 19–20  "All the time series have been subject to quality control, which caused the removal of those series, which showed multiple spikes or spectra that did not converge to zero at the highest frequencies."

The meaning of this sentence is unclear! What does it mean for a spectrum to "converge to zero" at the highest frequencies? Turbulence spectra decay as $k^{-5/3}$ in the inertial subrange …

Do you mean spectra displaying noise in the higher frequencies? Not falling off as $k^{-5/3}$, levelling off?

Please explain.

p. 4, l. 33 – p. 5, l. 4  There appears to be a conflict of notation between $C$ for the cospectrum and $C$ for the concentration of $CO_2$.

Eq. (1) and (2)  How did you calculate $\theta_{22}$, $\theta_{41}$ and $\theta_{80}$? From what instrument? Temperature profiles are sensitive to bias in the sensors: were the temperature sensors at these heights intercompared before deployment?

p. 5, l. 1–2  "and the standard deviation of the vertical wind component is $\sigma_w = \sum_{\tau} S_w$".

Wrong: the relationship is

$$\sigma_w^2 = \sum_{\tau} S_w.$$

Authors: check your calculations carefully to see if this is just a typo, or if you actually calculated (and are reporting) wrong values.

p. 5, l. 2–3     "Other variables, such as the Richardson number (Ri) and average horizontal wind speed (V) were calculated using the same data series used in the multiresolution decomposition."

Too vague: were the mean velocities from the sonics? Very important (see Main remarks above): from which sensors do the mean temperatures come?

p. 5, l. 22     Again, how were the $\theta$'s measured?

Section 3     Rename the section to indicate that it is about the comparison of two nights, one fully turbulent and the other intermittently turbulent. Suggestion: **Comparison of turbulence characteristics in a fully turbulent with and intermittently turbulent night.**

p. 5, l. 25–26     "The nocturnal flow at the site is characterized by the superposition of turbulent and non-turbulent fluctuations. In a fully turbulent night, such as 15 November 2015 (Fig. 1), there is a clear dominant wind direction at all levels."

Figure 1 does not show wind directions at the different levels. It is impossible to infer wind direction at each level from the figure.

p. 6, l. 3–8 and Table 1     "The most relevant difference between the two nights regards the magnitude of the turbulent mixing (Table 1). All relevant turbulence statistics are significantly larger on 15 November than on 14 November. The relative difference of the turbulence statistics between nights increases steadily in the vertical. As an example, TKE at 41 m is 3.4 times larger in the turbulent night than in the intermittent case, while at 80 m, TKE is 8.2 times larger in the turbulent night. Similar increases occur for the corresponding ratios of $\sigma_w$ and $u_*$ between the two nights."

The authors should reserve the symbol $u_*$ for a single value in each period, which should be the most representative for the friction between the flow above the canopy and the forest. Obviously this would be the value reported at 41 m. The others are "local" values of the kinematic momentum flux, and it would be more appropriate to write them as $\sqrt{-\overline{w'u'}}$. Same comment applies for $\theta_*$, etc..

Fig. 1-d, Fig. 2-d   The title $CO_2$ is missing from the left vertical axis.

p. 7, l. 10     "All quantities showed much larger variation across the levels in the intermittent night (Fig. 2). Furthermore, sporadic events of coupling occurred during bursts of intermittent turbulence (Fig. 2, shaded areas)."

The authors never explain the exact quantitive criterion for the identification of the shaded areas. It *appears* to be TKE, but they should give the quantitative criterion in the text.

**References**

J C. Kaimal. Turbulence spectra, length scales and structure parameters in the stable surface layer. *Boundary-Layer Meteorol.*, 4:289–309, 1973.

---

## Referee Comment (RC2) · Anonymous Referee #2 · 2 Oct 2017

The authors analyzed nighttime vertical fluxes of heat, water vapour, carbon dioxide, and ozone within and above a rainforest. They used multiresolution decomposition to determine the scales of atmospheric motions contributing to the vertical fluxes, focusing on low-frequency, non-turbulent fluctuations. I suggest rejection of the current manuscript because the authors failed to address a fundamental issue that no sonic anemometer can be aligned perfectly with the vertical direction perpendicular to the underlying surface. One may use a plumb bob to align sonic anemometer with the vertical direction, but an error of one or two degrees is expected over flat topography, and the error can be larger over slopes. One can use coordinate rotation techniques in the post-processing, but an uncertainty of two degrees still exists (e.g., Forken et al. 2004;

[Figure]

Vickers and Mahrt 2006). Aubinet et al. (2003) highlighted that "The $2^o$ offset would induce systematic errors on the vertical velocity up to 0.05 m s$^{-1}$ under typical stable conditions and up to 0.11 m s$^{-1}$ under near-neutral conditions. The resulting error in the vertical advection flux in the presence of a 10 $\mu$mol mol$^{-1}$ vertical $CO_2$ concentration difference may be as high as 5 $\mu$mol m$^{-2}$ s$^{-1}$". The errors in sonic coordinate system estimates would convert a few percent of variation in horizontal velocity components to variation in the vertical velocity component. On low-frequency, non-turbulent scales, horizontal velocity components are typically two orders of magnitude larger than the vertical velocity component. Consequently, the artificial variation induced by errors in sonic coordinate system estimates is at least comparable to the true variation in vertical fluxes on low-frequency, non-turbulent scales. Using eddy-covariance measurements to draw conclusions about low-frequency, non-turbulent vertical fluxes does not make sense unless the authors can distinguish true variation in vertical fluxes and artificial variation inherited from horizontal velocity components due to errors in the sonic coordinate system estimates. This fundamental issue must be resolved before the manuscript can go to more detailed review.

References

Aubinet, Marc, Bernard Heinesch, and Michel Yernaux. (2003). Horizontal and vertical CO2 advection in a sloping forest. Boundary-Layer Meteorology, 108(3): 397-417.

Foken, Thomas, et al. (2004). Post-field data quality control. Handbook of Micrometeorology. Springer Netherlands. 181-208.

Vickers, Dean, and L. Mahrt. (2006). Contrasting mean vertical motion from tilt correction methods and mass continuity. Agricultural and Forest Meteorology, 138(1): 93-103.

---

## Editor Comment (EC1) · G. Fisch (Editor) · 5 Oct 2017

I think that the paper has a scientific value and the comments of the reviewer 2 can (and must) be answered by the authors. I recomend to send them the reviewers comments (#1 and #2) to give the final answer about the quality of this MS

---

## Author Comment (AC1) · 13 Nov 2017

**Reply to reviewers**

**Reply to Reviewer 1**

We thank the reviewer for the careful analysis on our manuscript and for the overall positive opinion on our study. He/she raises very important issues, which have been addressed, largely improving the manuscript, in our opinion. Below, we answer to each of the points raised.

**Reviewer says:**

*On page 2, l. 21–23, the authors say:*

> Equivalent analyses focusing on scalar flux cospectra have not been presented as often. Sakai et al. (2001) and Finnigan et al. (2003) used cospectral similarity to conclude that low-frequency contribution could account for missing energy and $CO_2$ fluxes in their respective budgets, but neither study addressed how the cospectra varied across the canopy.

*later, on p. 2, l. 30–33, they say:*

> This result indicates that the exchange of scalars between the canopy and the atmosphere at night may occur at longer time scales than those traditionally used in the eddy covariance approach.

*and again, on p. 10, l. 10–15:*

> Our results support these findings, adding the information that the non-turbulent contribution may dominate the exchange of $CO_2$ and humidity from the interior of the canopy in very stable nights as well. It is likely that the same process affects other scalars, such as $O_3$ , whose concentrations are perturbed by intermittent events as shown in Fig. 4b.

*However, in the conclusions, they find that low-frequency components are important within the canopy, but that, above the canopy, it is the "turbulent scales" that contribute most of the flux. There seems to be a contradiction between the Introduction (and other parts of the manuscript) and the Conclusions. The introduction should not lead the reader to believe in a situation that will not be supported by the analysis.*

We thank the reviewer for raising this issue and for addressing it in great detail. There is certainly a lot of confusion regarding this matter in the original version of the manuscript. Trying to be as simple as possible, what we meant to say is that:

- Above the canopy, turbulent exchange is always important;
- Within the canopy, nonturbulent exchange is always important;
- In very stable conditions, the nonturbulent contribution increases above the canopy as well. In such conditions, both turbulent and non-turbulent exchange become nearly as important.

Now, to correct the confusion, the following portions have been altered in the text:

In the original version (p. 2, l. 29):

> They also found that sensible heat flux cospectra within the canopy peaked at longer time scales, again similar to those of the non-turbulent maxima of horizontal TKE above the canopy. This result indicates that the exchange of scalars between the canopy and the atmosphere at

night may occur at longer time scales than those traditionally used in the eddy covariance approach.

**Has been replaced with (changes in red):**

They also found that sensible heat flux cospectra within the canopy peaked at longer time scales, again similar to those of the non-turbulent maxima of horizontal TKE above the canopy. Their results indicate that the exchange of scalars within the canopy at night may occur at longer time scales than those traditionally used in the eddy covariance approach. At very stable conditions, such long scales may also contribute to the total exchange between the canopy and the atmosphere.

**In the original version (p. 10, l. 9):**

Santos et al. (2016) found that the time scales of horizontal turbulent velocity fluctuations within an Amazonian rain forest canopy (at a different site) approach those of the non-turbulent maximum above the forest. They also found that the dominant time scales of the vertical velocity fluctuations and sensible heat flux within the canopy are shifted towards larger values than those above it. Our results support these findings, adding the information that the non-turbulent contribution may dominate the exchange of $CO_2$ and humidity from the interior of the canopy in very stable nights as well.

**Has been replaced with (changes in red):**

Santos et al. (2016) found that the dominant time scales of the vertical velocity fluctuations and sensible heat flux within an Amazonian rain forest canopy (at a different site) are shifted towards larger values than those above it. Our results support these findings, adding evidence that the exchanges of $CO_2$ and humidity within the canopy are also dominated by non-turbulent contribution at very stable nights.

**In the original version (p. 17, l. 4, at the conclusion section):**

The majority of the fluxes just above the canopy (41-m level, in this case) happens through turbulent exchange. Although no relevant systematic low-frequency contribution to the total fluxes has been found at 41 m for any scalar analyzed (Fig. 7), this result only holds for the average spectra, with appreciable variability among cases, especially in the most stable cases.

**Has been replaced with (changes in red, last sentence removed):**

Turbulent exchange is always important just above the canopy (41-m level, in this case) but, in very stable nights the non-turbulent contribution has to be also considered.

**Reviewer says:**

*"Bulk" Richardson numbers are used, but these are sensitive to velocity and, most of all, temperature systematic errors between the sensors. Because several analyses are dependent on these Richardson numbers, their reliability must be assessed quantitatively. Have the sensors been intercompared?*

*In the worst case (no intercomparison, no calibration), a thorough sensitivity analysis must be made of the effects of the temperature (and wind) systematic errors on those Richardson numbers and in the analyses involving them. The reported accuracies for the sensors (assuming optimistically that they did not drift) can be used as a basis for this. The uncertainty introduced by those errors results should then be displayed graphically in all analyses regarding the Richardson number.*

Temperature has been measured by a profile of thermohygrometers and by the sonic anemometers, as well. The thermohygrometers have been intercompared, while each sonic temperature has been compared to the closest thermohygrometer. These informations have been added to the manuscript.

**Reviewer says:**

*Turbulent bursts: the criterion for identifying the turbulent bursts and defining the shaded regions in Fig. 2 should be made clear (quantitavely).*

It has been assumed that a turbulent burst occurred whenever $\sigma_w > 0.15 \, m \, s^{-1}$ . This information has been added to the manuscript.

**Reviewer says:**

*Text starting on p. 11, l. 5, says*

> Sun et al. (2012) found two regimes of nocturnal turbulence, distinguished by the turbulent kinetic energy (TKE) dependence on the mean wind speed. The fully turbulent regime, typically associated with weakly stable conditions, happens for mean wind speeds larger than a height dependent threshold and is characterized by TKE that steadily increases with wind speed. The other regime, associated with very stable conditions, has reduced turbulence intensities, which are very weakly dependent on the mean wind speed. Dias-Júnior et al. (2017) observed the two regimes above the forest at a site in the southwestern Amazon, finding that each is associated with an independent lognormal frequency distribution of quantities such as the turbulence dissipation rate. For the turbulent night of 15 November 2015 (Fig. 8, crosses), the levels of 41 and 55 m remained in the large wind speed regime for the whole period, while the two different regimes could be observed only at the 80-m level. On the intermittent night, on the other hand (Fig. 8, triangles), both regimes could be observed at all levels. Moreover, the connection intervals, given by shaded areas in Fig. 2, are generally in the large wind speed regime both at 41 m and 55 m (Fig. 8, filled triangles), while the decoupled periods are in most cases in the weak wind regime (Fig. 8, open triangles). **This is an important result, because it indicates that the intermittent bursts of turbulence observed above the canopy are intense enough to cause a regime transition. It means that, during these events, there is likely full vertical coupling over the vertical extent of the SBL (which is, at this time, shallower than 80 m). Therefore, scalars that are emitted from the canopy may be able to escape to higher levels in the boundary layer, as suggested by the episodic mixing of $CO_2$ and $O_3$ above 70 m shown in Fig. 4.**

*(my emphasis). But high turbulent fluxes above the canopy during the the bursts of turbulent activity are already clearly displayed in Fig 5. The "full coupling" is none other than the relatively high (absolute) values of the fluxes themselves. Given that the fluxes are there, the scalars have already "escaped" the canopy. Therefore, the reasoning in the bold-face text above seems to be rather circular, and nothing new seems to arise from this discussion. Moreover, if the criterion for identifying the bursts was TKE (as I suspect), then it is inevitable that this will be reflected in higher TKE values in Fig. 8. It appears to me that the definition of the bursts and the regime classification in Fig. 8 are one and the same, and that there is nothing to be added here. I strongly suggest deleting this whole passage.*

It is possible that an intermittent event is not intense enough to cause regime transition. In this case, they are local, and they lie at the weak wind side of the *V* vs $V_{TKE}$ diagram (despite the TKE enhancement). However, we understand that this type of intermittence has never been considered in this study, and that its abrupt introduction to the manuscript may be rather confusing to the reader. Therefore, we agree with the reviewer´s suggestion of removing this passage, which we will keep to a specific study on regime transitions.

**Reviewer says:**

*Sections 4 and 5 seem to use all the data from the 15 usable nights. Because the previous section focused strongly on the comparison of the nights of Nov 14 and 15, I had a hard time (at first reading) realizing this. I suggest that both the title and the introduction of each of these sections reinforces the information that, now, data from all 15 nights are being analyzed.*

Title of section 4 has been changed to "Mean spectra and cospectra over the 15 nights", while that of section 5 has been changed to "Dependence on stability over the 15 nights". Furthermore first sentence of section 4 now reads "Fig. 7 shows the spectra and cospectra of the turbulent fluctuations and fluxes averaged over the entire period of 15 nights". First sentence of section 5 now reads "The comparison of the fluxes determined with 5-min and 109-min time windows and of their stability dependence for the entire period of 15 nights provides interesting information on the scalar exchange processes within and above the canopy"

**Reviewer says:**

*(p. 12, l. 10): "Figure 9 shows the spectra and cospectra of the turbulent fluctuations and fluxes averaged over the entire period".*
*Particularly in stable conditions, there is a strong shift of the spectra towards the higher frequencies with increasing stability (Kaimal, 1973). There is no equation describing how the spectra were "averaged", but there should be. The simplest approach (which I suspect is being used here) is to average per frequency. But then, because frequency depends on stability, different stabilities and their spectral densities are being averaged together. The consequences are far from clear to me, and this procedure should not be done without careful justification. Remember, if y = f (x) and f is nonlinear, then $\overline{y} \neq f(\overline{x})$ in general. It is not clear how the fluxes reported in Sect. 5 were calculated. Are they bin averages? Do they come from the integration of the mean spectra? If $F^{(i)}_{wa}$ is the flux from the $i^{th}$ cospectrum, and if $F_{wa,mean}$ is the flux from the mean cospectrum (as depicted in Fig. 9), how do $(1/n)\sum_{i=1}^{n} F^{(i)}_{wa}$ and $F_{wa,mean}$ compare? In this sense, how valuable and correct are the conclusions derived from Fig. 9?*

This is an important issue, we thank the reviewer for raising it. First, he/she is correct that we are averaging by frequency. In that sense, Fig. 9 is being affected by this problem, which may cause the spectra to "spread" in the horizontal. We do not think this is a problem, as the paper does not focus on scaling issues but, rather, on reporting physical processes, the contrast between turbulent and nonturbulent exchange. These are clear in Fig. 7, despite some spreading over the time scale axis. Nevertheless, to make it clear, we added an explanation at the caption of Figs. 6, 7, 9, 13 and 14 that they show averages by frequency. ("in all panels, averages are performed over each time scale").
Besides, in figs. 13 and 14, the two classes represent different stabilities, and it can be seen that the scales of the turbulent fluxes vary much less between classes of stability than does the magnitude of the nonturbulent contribution to the fluxes.

Finally, all flux estimates from cospectra have been done for each series separately, so that their average shown are $(1/n)\sum_{i=1}^{n} F_{wa}^{(i)}$ rather than $F_{wa,mean}$ (using the reviewer notation).

This has been now properly stated in the manuscript, to avoid confusion from the readers: "Variances and fluxes with a 109-min long time average have been obtained from the integration of the respective multiresolution spectra and cospectra for each series separately, and then averaged, if appropriate."

**Reviewer says:**

*p. 4, l. 15–16: "Since the different levels of flow structures are analyzed simultaneously, only the data when all levels were available was used." This should be: ". . . Since the different levels of flow structures are analyzed simultaneously, only the data when all levels were available **were** used".*

Done.

**Reviewer says:**

*p. 4, l. 19–20: "All the time series have been subject to quality control, which caused the removal of those series, which showed multiple spikes or spectra that did not converge to zero at the highest frequencies."*
*The meaning of this sentence is unclear! What does it mean for a spectrum to "converge to zero" at the highest frequencies? Turbulence spectra decay as $k^{-5/3}$ in the inertial subrange . . . Do you mean spectra displaying noise in the higher frequencies? Not falling off as $k^{-5/3}$, levelling off?*
*Please explain.*

We simply meant "multiresolution spectra that displayed noise at the shortest time scales".

**Reviewer says:**

*p. 4, l. 33 – p. 5, l. 4 There appears to be a conflict of notation between C for the cospectrum and C for the concentration of $CO_2$.*

All occurrences of "C" alone referred to cospectrum.

**Reviewer says:**

*Eq. (1) and (2) How did you calculate $\theta_{22}$, $\theta_{41}$ and $\theta_{80}$? From what instrument? Temperature profiles are sensitive to bias in the sensors: were the temperature sensors at these heights intercompared before deployment?*

They have been determined from the sonic temperatures, which have been compared to the closest thermohygrometer, as explained in the reply to the second comment from the reviewer, above.

**Reviewer says:**

*p. 5, l. 1–2: "and the standard deviation of the vertical wind component is $\sigma_w = \sum_\tau S_w$ ".*

*Wrong: the relationship is $\sigma_w^2 = \sum_\tau S_w$*

*Authors: check your calculations carefully to see if this is just a typo, or if you actually calculated (and are reporting) wrong values.*

It was just a typo, thank you for noticing it!

**Reviewer says:**

*p. 5, l. 2–3: "Other variables, such as the Richardson number (Ri) and average horizontal wind speed (V ) were calculated using the same data series used in the multiresolution decomposition."*

*Too vague: were the mean velocities from the sonics? Very important (see Main remarks above): from which sensors do the mean temperatures come?*

Yes, both came from the sonic data. We used this phrasing to state that the same time intervals were chosen as those used for the multiresolution decomposition. Sentence now reads: "Other variables, such as the Richardson number (Ri) and average horizontal wind speed (V ) were calculated using the same time intervals used in the multiresolution decomposition."

**Reviewer says:**

*p. 5, l. 22 Again, how were the θ's measured?*

From the sonic data, which have been compared to the thermohygrometers. This information has been added to the manuscript.

**Reviewer says:**

*Section 3:  Rename the section to indicate that it is about the comparison of two nights, one fully turbulent and the other intermittently turbulent. Suggestion: Comparison of turbulence characteristics in a fully turbulent with and intermittently turbulent night.*

Done.

**Reviewer says:**

*p. 5, l. 25–26 "The nocturnal flow at the site is characterized by the superposition of turbulent and non-turbulent fluctuations. In a fully turbulent night, such as 15 November 2015 (Fig. 1), there is a clear dominant wind direction at all levels."*

*Figure 1 does not show wind directions at the different levels. It is impossible to infer wind direction at each level from the figure.*

Figure direction can be inferred from the signals of the two horizontal wind components. To clarify this matter, wind direction has been associated to the sign change of the components at another sentence (this association has already been made once, at the previous sentence): "In contrast, during the intermittent night of 14 November 2015 (Fig. 2), there is no dominant wind direction at any level above the canopy, as both horizontal components switch sign many times along the night".

**Reviewer says:**

*p. 6, l. 3–8 and Table 1 "The most relevant difference between the two nights regards the magnitude of the turbulent mixing (Table 1). All relevant turbulence statistics are significantly larger on 15 November than on 14 November. The relative difference of the turbulence statistics between nights increases steadily in the vertical. As an example, TKE at 41 m is 3.4 times larger in the turbulent night than in the intermittent case, while at 80 m, TKE is 8.2 times*

*larger in the turbulent night. Similar increases occur for the corresponding ratios of $\sigma_w$ and $u_*$*
*between the two nights."*

*The authors should reserve the symbol $u_*$ for a single value in each period, which should be the most representative for the friction between the flow above the canopy and the forest. Obviously this would be the value reported at 41 m. The others are "local" values of the kinematic momentum flux, and it would be more appropriate to write them as $\sqrt{\overline{w'u'}}$. Same comment applies for $\theta_*$, etc..*

The suggestion has been accepted.

**Reviewer says:**

*Fig. 1-d, Fig. 2-d The title $CO_2$ is missing from the left vertical axis.*

Done.

**Reviewer says:**

*p. 7, l. 10*

*"All quantities showed much larger variation across the levels in the intermittent night (Fig. 2). Furthermore, sporadic events of coupling occurred during bursts of intermittent turbulence (Fig. 2, shaded areas)."*

*The authors never explain the exact quantitive criterion for the identification of the shaded areas. It appears to be TKE, but they should give the quantitative criterion in the text.*

This issue has already been addressed in the third comment from the reviewer, above.

**Reply to Reviewer 2**

Despite the rejection suggestion, with which we do not agree, we thank the reviewer for raising the issue of coordinate rotation. This is, as the reviewer points out, a key issue in a study of this kind and it had not been properly clarified in the original version. Below is our answer.

**Reviewer says:**

The authors analyzed nighttime vertical fluxes of heat, water vapour, carbon dioxide, and ozone within and above a rainforest. They used multiresolution decomposition to determine the scales of atmospheric motions contributing to the vertical fluxes, focusing on low-frequency, non-turbulent fluctuations. I suggest rejection of the current manuscript because the authors failed to address a fundamental issue that no sonic anemometer can be aligned perfectly with the vertical direction perpendicular to the underlying surface. One may use a plumb bob to align sonic anemometer with the vertical direction, but an error of one or two degrees is expected over flat topography, and the error can be larger over slopes. One can use coordinate rotation techniques in the post-processing, but an uncertainty of two degrees still exists (e.g., Forken et al. 2004; Vickers and Mahrt 2006). Aubinet et al. (2003) highlighted that "The 2° offset would induce systematic errors on the vertical velocity up to 0.05 m s$^{-1}$ under typical stable conditions and up to 0.11 m s$^{-1}$ under near-neutral conditions. The resulting error in the vertical advection flux in the presence of a 10 μmol mol$^{-1}$ vertical $CO_2$ concentration difference may be as high as 5 μmol m$^{-2}$ s$^{-1}$ ". The errors in sonic coordinate system estimates would convert a few percent of variation in horizontal velocity components to variation in the vertical velocity component. On low-frequency, non-turbulent scales, horizontal velocity components are typically two orders of magnitude larger than the vertical velocity component. Consequently, the artificial variation induced by errors in sonic coordinate system estimates is at least comparable to the true variation in vertical fluxes on low-frequency, non-turbulent scales. Using eddy-covariance measurements to draw conclusions about low-frequency, non-turbulent vertical fluxes does not make sense unless the authors can distinguish true variation in vertical fluxes and artificial variation inherited from horizontal velocity components due to errors in the sonic coordinate system estimates. This fundamental issue must be resolved before the manuscript can go to more detailed review.

In the present study, double rotation has been applied to each individual time series, following the suggestions from Acevedo and Mahrt (2010). Besides, only scalar, rather than momentum fluxes are considered. We believe that those results provide strong foundation for us to use this technique in the present study. However, they certainly need to be properly mentioned in the manuscript, and they were not in the original version. Therefore, the following paragraph has been added to section 2:

> Acevedo and Mahrt (2010) used the multiresolution decomposition to analyze vertical profiles of the non-turbulent component of sensible heat fluxes. They found that systematic and organized profiles, whose inclusion contributes for the closure of the nocturnal temperature budget near the surface, are only found when the double wind rotation (Tanner and Thurtell, 1969) is applied to each time series analyzed, separately. They claim that this is more suitable for such analysis than other coordinate rotation procedures, such as globally directionally dependent methods (Lee, 1998; Mahrt et al., 2000; Paw U et al., 2000) because "…*the measured vertical motion on times scales greater than 5 h may be sufficiently weak and unreliable that the elimination of larger-scale variations of vertical motion through coordinate rotation improves the calculation*". For these reasons, the double rotation was applied to each 109-min time series separately.

Finally, we would like to stress that although a very important part of the paper, the contrast between turbulent and nonturbulent modes of the flow is not all that is addressed in the study. Furthermore, not only fluxes ("exchange") are analyzed, as scalar profiles and power spectra are also investigated. For this reason, and to take the focus of the study away from this

matter, we are also changing the paper title to "Nighttime wind and scalar variability within and above an Amazonian canopy".

[revised manuscript text omitted]

---

## Editor Comment (EC2) · G. Fisch (Editor) · 23 Nov 2017

On my oppinion, the authors did most of the corrections/sugestions made by the reviewers. In particularly, they have addressed very efficiently the major point raised by reviewer #2 about coordinates correction on high frequency data. This paper will bring new insights about the turbulence over a tropical forest in cental Amazonia.